# The SIAH2-NRF1 axis spatially regulates tumor microenvironment remodeling for tumor progression

Biao Ma[1], Hongcheng Cheng[1], Chenglong Mu[1], Guangfeng Geng[1], Tian Zhao[1], Qian Luo[1], Kaili Ma[1]
Rui Chang[1], Qiangqiang Liu[1], Ruize Gao[1], Junli Nie[1], Jiaying Xie[1], Jinxue Han[1], Linbo Chen[1], Gui Ma[1],
Yushan Zhu[1] & Quan Chen[1,2]

The interactions between tumor cells with their microenvironments, including hypoxia, acidosis and immune cells, lead to the tumor heterogeneity which promotes tumor progression. Here, we show that SIAH2-NRF1 axis remodels tumor microenvironment through regulating tumor mitochondrial function, tumor-associated macrophages (TAMs) polarization and cell death for tumor maintenance and progression. Mechanistically, low mitochondrial gene expression in breast cancers is associated with a poor clinical outcome. The hypoxia-activated E3 ligase SIAH2 spatially downregulates nuclear-encoded mitochondrial gene expression including *pyruvate dehydrogenase beta* via degrading NRF1 (Nuclear Respiratory Factor 1) through ubiquitination on lysine 230, resulting in enhanced Warburg effect, metabolic reprogramming and pro-tumor immune response. Dampening NRF1 degradation under hypoxia not only impairs the polarization of TAMs, but also promotes tumor cells to become more susceptible to apoptosis in a FADD-dependent fashion, resulting in secondary necrosis due to the impairment of efferocytosis. These data represent that inhibition of NRF1 degradation is a potential therapeutic strategy against cancer.

[1] State Key Laboratory of Medicinal Chemical Biology, Tianjin Key Laboratory of Protein Sciences, College of Life Sciences, Nankai University, Tianjin 300071, China. [2] State Key Laboratory of Membrane Biology, Institute of Zoology, Chinese Academy of Sciences, Beijing 100101, China. These authors contributed equally: Biao Ma, Hongcheng Cheng. These authors jointly supervised this work: Biao Ma, Yushan Zhu, Quan Chen. Correspondence and requests for materials should be addressed to B.M. (email: biaoma@mail.nankai.edu.cn) or to Y.Z. (email: zhuys@nankai.edu.cn) or to Q.C. (email: chenq@ioz.ac.cn)

Mitochondria, the central platform of cellular metabolism involving oxidative phosphorylation, tricarboxylic acid (TCA) cycle, and fatty acid β-oxidation[1,2], supply most of the cellular ATP and various metabolic intermediates needed for the cellular energy demands, building blocks of cellular biomass and signal transductions. In order to maintain proper cellular functions, the balance of mitochondrial mass is strictly regulated, including mitochondrial biogenesis and turnover process[1]. Dysregulation of mitochondrial homeostasis may cause improper mitochondrial function, leading to altered cell morphology and function or even diseases, such as cancer. Decades ago, Otto Warburg observed that tumors utilize glycolysis for energy production even in the presence of sufficient oxygen, which implied that dysfunctional mitochondria might support tumorigenesis[3–8]. Consistently, low mtDNA copy number has also been observed in various types of cancer[9], which is supportive of mitochondrial dysfunction within tumors. However, it was also observed that tumor growth actually requires functional mitochondria[10–13]. Therefore, the roles of mitochondria in tumorigenesis appear to be paradoxical and the hypothesis from Warburg remains contentious. Despite numerous of evidence showing that mutations in mtDNA, low mtDNA copy number and respiratory defects are commonly seen in various types of cancer[14], direct evidence linking tumorigenesis and mitochondrial biogenesis remain missing.

In addition, tumor cells are situated in highly heterogeneous microenvironments, both in cellular composition and metabolic profiling[15]. Tumor metabolic heterogeneity is also believed to play a role in chemo-resistance, distant metastasis and tumor recurrence, resulting in poor clinical outcome[15]. However, little is known about how mitochondria, as the most important organelle involved in metabolism within the cell, respond to tumor microenvironmental cues and how they regulate the tumor metabolic heterogeneity. Understanding the role of mitochondrial regulation in the context of tumor microenvironment is essential to decipher the molecular basis of tumor progression.

We therefore tried to reveal the difference of nuclear-encoded mitochondrial gene (NEMG) expression between normal tissues and breast cancer tissues, and the correlation between NEMG expression and clinical outcome in breast cancer patients. Our results also identified that mitochondria are spatially organized in response to tumor hypoxia through the degradation of NRF1 by the hypoxia-induced E3 ligase SIAH2, resulting in mitochondrial heterogeneity which consequentially potentiates metabolic heterogeneity. For instance, productions of lactate and prostaglandin E2 (PGE2) were found increased under hypoxia. Furthermore, we also identified that Fas-associated protein with death domain (FADD), which encodes a core component of death-inducing signaling complex (DISC) during apoptosis, is a NRF1 target gene and its expression is reduced in response to hypoxia. Ablation of NRF1 degradation enhances tumor cell apoptosis in a FADD-dependent manner, which induces secondary necrosis due to insufficient clearance of the apoptotic cell caused by the lack of polarized TAMs. Thus, these results reveal a series of intrinsic and extrinsic cellular reactions during tumor progression and demonstrate the importance of these processes in tumor maintenance.

## Results

### NEMG expression negatively correlates with clinical outcomes.
Using large-scale gene transcription profiling data from the Gene Expression Omnibus (GEO) database, we identified that breast tumors had significantly reduced transcription of genes enriched in gene sets: precursor metabolites generation, fatty acid oxidation, and energy derivation through oxidation (Fig. 1a). Most of these genes are NEMGs, and the results are indicative of mitochondrial reprogramming during carcinogenesis. Tissue microarray analysis using an antibody against the mitochondrial marker Prohibitin revealed that the mitochondrial proteins were significantly reduced in human breast tumor tissues (Fig. 1b and Supplementary Table 1), which further validated our results from the bioinformatics analysis. Furthermore, the reduction of Prohibtin was found to be positively correlated with clinical stage (Fig. 1c, d and Supplementary Table 2). Clinical data also showed that decreased transcription of several mitochondrial genes, such as *PHB*, *POLG2*, *MPC2*, *PDHB*, *NDUFA3* and *CPT2* significantly correlated with lower relapse-free survival rates (Fig. 1e).

Intriguingly, immunostaining showed that the hypoxia marker GLUT1 and the mitochondria marker TIMM23 had an inverse expression pattern in spontaneous breast tumor tissues (Fig. 1f). These results indicate that tumor tissues are highly complex and contain oxygen level-associated mitochondrion-rich regions and mitochondrion-poor regions, which is termed as mitochondrial heterogeneity. Since hypoxia is a common feature of solid tumors and also a prognostic marker of malignancy[16], we then asked how this mitochondrial distribution pattern is formed in response to tumor hypoxia and what its physiological functions may be. Previous researches have shown that HIFs play a role in regulating mitochondrial mass and function under hypoxic conditions by upregulating HIFs target genes *NIX* and *BNIP3*, to induce mitophagy[17,18], or *PDK*[19] and *COX IV2* (ref. [20]) to inhibit TCA cycle and electron transport chain. However, we found that the expression of many genes involved in normal mitochondrial function was markedly inhibited by hypoxia in MDA-MB-231 breast cancer cells (Fig. 1g), raising the question that what is the biological significance of hypoxia-induced transcriptional regulation of NEMG expression.

We therefore hypothesized that hypoxia, as a tumor microenvironmental factor[16], may affect clinical outcomes in patients by inhibiting NEMG expression. Clinical patient data were analyzed to testify the hypothesis. The median values of the transcriptional levels of those conventional hypoxia-induced genes, including *VEGF*, *LOX*, *ENO1*, *GLUT1*, *HMOX1* and *PLAUR* were independently defined as the hypoxia parameter thresholds[16], and the samples were grouped into normoxia (<threshold) and hypoxia (>threshold) accordingly. Strikingly, NEMGs were consistently enriched in the normoxic samples, which were associated with significantly higher disease relapse-free survival rates in breast cancer patients (Fig. 1h). These data indicate that hypoxia-mediated reduction of NEMG expression may associate with a poor outcome in breast cancer patients, which is consistent with our previous results (Fig. 1b–d).

### Hypoxia regulates NEMG expression through NRF1.
Normal cellular homeostasis requires a balanced mitochondrial pool which is regulated by both mitochondrial biogenesis and turnover processes[1]. However, how microenvironmental cues regulate this balance is largely uncertain. We first checked the mitochondrial proteins under prolonged hypoxia and the results showed that a dramatic reduction in the levels of mitochondrial proteins, including NRF1 (one of the master regulators of mitochondrial biogenesis[21,22]), was triggered in the MDA-MB-231 cells in a time-dependent manner (Fig. 2a and Supplementary Fig. 1a). Similar results were also observed in HeLa cells (Supplementary Fig. 1b). We further showed that the reduction of these proteins was accompanied by transcriptional downregulation of the corresponding genes under hypoxia with an exception of NRF1 (Fig. 2b and Supplementary Fig. 2a), which supports our notion that reduced NEMG expression may be due to the downregulation of NRF1. Meanwhile, mouse embryonic fibroblast cells

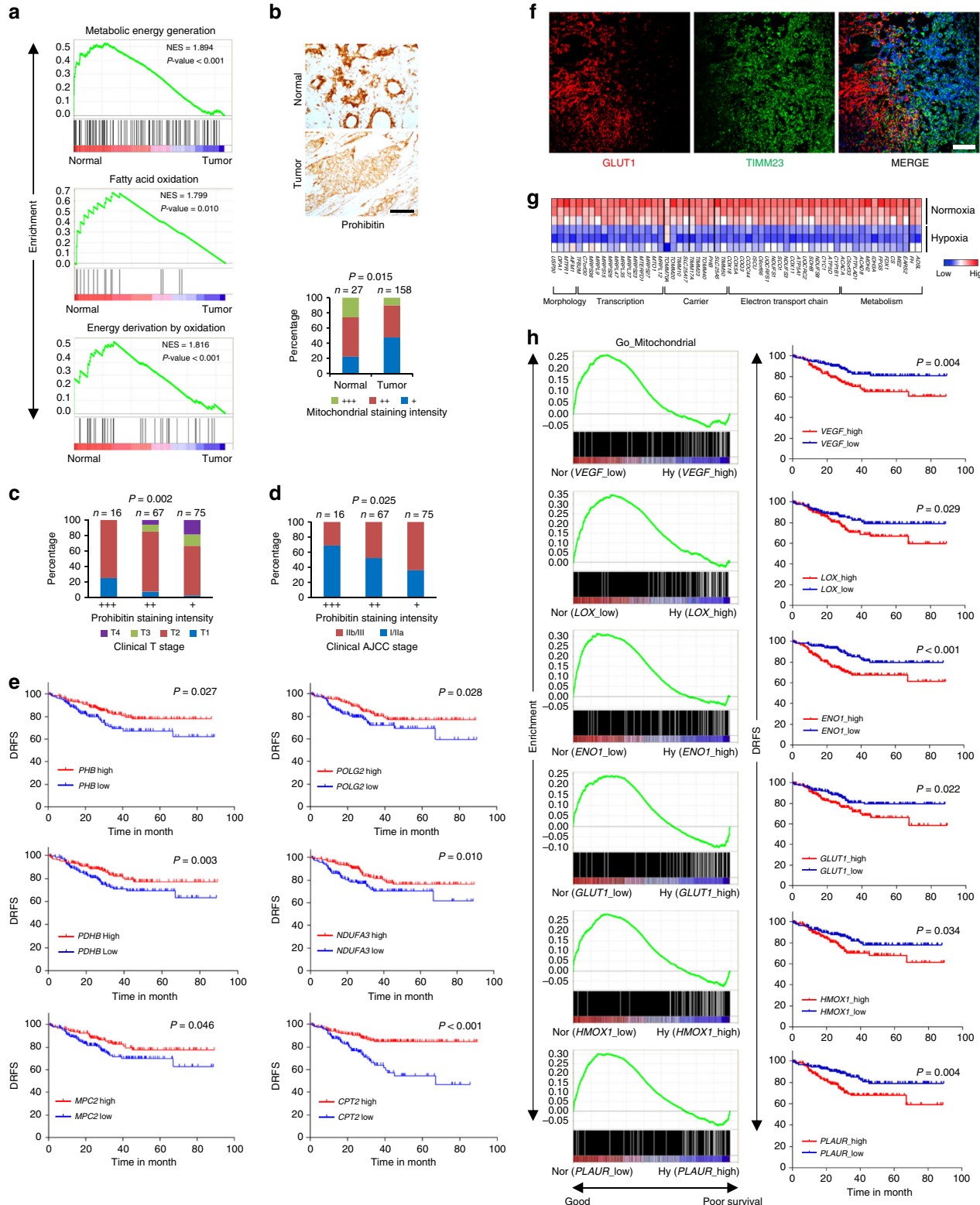

(MEFs) depleted of *ATG5*, a known regulator of general autophagy and mitophagy[23,24], showed similar reduction of mitochondria markers seen in MDA-MB-231 and HeLa cells under hypoxia (Supplementary Fig. 1c-e), which exclude the possibility that the persistent reduction is due to autophagic degradation, while may be the result of decreased NEMG expression.

To further validate the impact of NEMG expression on mitochondrial proteins under hypoxia, cycloheximide (CHX) was

used to block protein synthesis. If the mitochondrial degradation process is stimulated by hypoxia, mitochondrial marker proteins should decrease dramatically under hypoxia; but if NEMG expression plays a dominant role in regulating the mitochondrial proteins under hypoxia, the markers should relatively remain constant during hypoxia due to ceased protein synthesis. The results turned out to support the latter explanation, i.e. the mitochondrial proteins remained constant under hypoxia after

**Fig. 1** Nuclear-encoded mitochondrial gene expression negatively correlates with clinical outcome in breast cancer patients. **a** GO (Gene Ontology) enrichment analysis in normal and breast cancer tissues from dataset GSE15852, with a false discovery rate (FDR) of <25%. **b** Top: immunohistochemical staining of Prohibitin in a representative normal breast tissue sample and breast cancer tissue sample from the tissue microarray. Scale bars, 50 μm. Brown color indicates positive immune reaction. Bottom: graph showing the intensity of Prohibitin staining in 27 normal and 158 tumor tissue samples. Statistical significance was determined by $\chi^2$ test. **c**, **d** Statistical analysis of correlations between Prohibitin staining intensity and clinical T stage (**c**) or clinical AJCC stage (**d**). Statistical significance was determined by $\chi^2$ test. **e** Analysis of the correlations between disease relapse-free survival rate (DRFS) and the expressions of indicated genes in breast cancer patients from dataset GSE25055. **f** Mouse spontaneous breast tumor tissues were stained with anti-GLUT1 (red) and anti-TIMM23 (green) antibodies together with DAPI (blue). Scale bars, 50 μm. **g** Heat map of the expression of mitochondrial genes in MDA-MB-231 cells cultured under normoxia and hypoxia (dataset GSE18494). **h** Analysis of the correlations between DRFS and GO enrichment in mitochondrial genes in breast cancer patients from dataset GSE25055

inhibition of new mitochondrial protein synthesis by CHX treatment (Fig. 2c and Supplementary Fig. 2b).

To address the role of NRF1 in regulating mitochondrial proteins in response to hypoxia, a stable NRF1-knockdown line was generated in MDA-MB-231 cells and used to perform the same experiments. As expected, the mitochondrial marker levels remained constant under hypoxia (Fig. 2d and Supplementary Fig. 2c). This mechanism seems to be a general rule in regulating mitochondrial proteins under hypoxia among cell lines derived from different sub-types of breast cancer. All these cell lines showed that knockdown of NRF1 under normoxia mimicked the mitochondrial phenotypes under hypoxia, which is shown as reduced mitochondrial proteins and less sensitivity to hypoxia compared with wild-type cells (Fig. 2e). This further validated our hypothesis that NRF1-dependent downregulation of NEMG is responsible for the hypoxia-induced reduction of mitochondrial proteins, rather than enhanced mitochondrial turnover. Taken together, these data demonstrate that NRF1 is a critical switch that regulates mitochondrial proteins under hypoxia and imply that hypoxic suppression of NRF1 may be responsible for the formation of mitochondrial heterogeneity in tumors (Fig. 1f).

**SIAH2 promotes NRF1 ubiquitination and degradation.** Since NRF1 is critical for maintaining NEMG expression, we would like to clarify the molecular mechanism underlying the dramatic reduction of NRF1 induced by hypoxia. We found that the proteasome inhibitor MG132 but not the autophagy inhibitor Bafilomycin A1 (BafA1), dramatically blocked hypoxia-induced NRF1 degradation (Fig. 3a), indicating that the degradation of NRF1 mainly occurs through the proteasomal pathway. Accordingly, NRF1 was also found significantly polyubiquitinated under hypoxic conditions (Fig. 3b). To identify the E3 ligase that potentially ubiquitinates NRF1, several ubiquitin E3 ligases previously reported to be involved in hypoxia response[25–29] were screened and SIAH2 was identified to be responsible for the stability of NRF1 (Fig. 3c). Further analysis showed that SIAH2 promotes NRF1 degradation through the proteasomal pathway in a dosage-dependent manner requiring its E3 ligase activity (Supplementary Fig. 3a–d).

Direct interaction between SIAH2 and NRF1 was shown via co-immunoprecipitaion and GST-pulldown assays (Fig. 3d and Supplementary Fig. 3e, f), and the critical binding region on NRF1 was identified as the amino acids 109–300 (Supplementary Fig. 3g, h). In vivo and in vitro ubiquitination assays further confirmed that NRF1 is a direct substrate of SIAH2 (Fig. 3e, f). SIAH2 is known to promote the auto-ubiquitination and degradation under normoxia, which can be reversed by hypoxia via promoting the binding with its substrates[30–34]. Consistently, we found that the interactions between NRF1 and SIAH2 were enhanced under hypoxia (Fig. 3g). Moreover, SIAH2 depletion by CRISPR knockout stabilized NRF1 under normoxic condition (Fig. 3h), and even completely abolished hypoxia-induced ubiquitination (Fig. 3i) and degradation (Fig. 3h) of

NRF1. These results were identical to those obtained from SIAH2-knockdown cells (Supplementary Fig. 3i, j), indicating that SIAH2 is the major endogenous E3 ligase responsible for NRF1 stability regulation. Next, we checked the expression level of NRF1 and SIAH2 in clinical samples and found that NRF1 and SIAH2 were abundant in normal and breast cancer tissues, respectively (Fig. 3j and Supplementary Table 3, 4). This suggests that breast tumorigenesis may associate with the downregulation of NRF1. Furthermore, reduction of NRF1 was also found significantly correlated with cancer progression (Supplementary Table 5). However, clinical data showed no significant difference in NRF1 transcriptional levels between normal and breast cancer tissues (Supplementary Fig. 4), supporting our notion that the regulation of NRF1 is mainly at the posttranslational level in patients.

**NRF1 is ubiquitinated on Lysine 230 by SIAH2.** As NRF1 is a substrate of SIAH2, we wanted to determine the exact site responsible for SIAH2-mediated ubiquitination on NRF1 in response to hypoxia. All 17 lysine residues of NRF1 were mutated to arginine one by one and lysine 230 was identified as the key site (Fig. 4a). K230R was completely resistant to SIAH2-mediated degradation (Fig. 4b) and ubiquitination in vivo (Fig. 4c) and in vitro (Fig. 4d). To exclude the possibility that K230R compromised the interaction between NRF1 and SIAH2, we performed co-immunoprecipitation experiments, which showed that K230R did not affect the NRF1/SIAH2 interaction (Fig. 4e). As expected, K230R was also completely resistant to hypoxia-induced ubiquitination (Fig. 4f) and degradation (Fig. 4g). These data provide further evidence supporting that lysine 230 is the only site in NRF1 for SIAH2-mediated ubiquitination under hypoxia.

**SIAH2 regulates NEMG expression through NRF1.** Based on our previous results, we conclude that hypoxia, as a micro-environmental cue, promotes NRF1 degradation through SIAH2. Considering NRF1 as a master regulator of NEMG expression[19,20], we speculated that SIAH2 may be the executioner of inhibiting NEMG expression under hypoxia. Indeed, SIAH2 deficiency dramatically stabilized mitochondrial markers under normoxic conditions and almost completely restrained the hypoxia-induced reduction of these markers (Fig. 5a). However, when NRF1 was simultaneously knocked down in SIAH2-deficient cells, these phenotypes caused by loss of SIAH2 were completely reversed (Fig. 5a). Further analysis showed that the hypoxia-induced downregulation of NEMG expression was abrogated in SIAH2$^{-/-}$ cells, which was reversed by depletion of NRF1 (Fig. 5b). These results indicate that the SIAH2-NRF1 axis is involved in hypoxia-induced downregulation of NEMG expression. These data were also consistent with our bioinformatics analysis showing that mitochondrial genes were significantly enriched in SIAH2$^{-/-}$ tissues (Supplementary Fig. 5a).

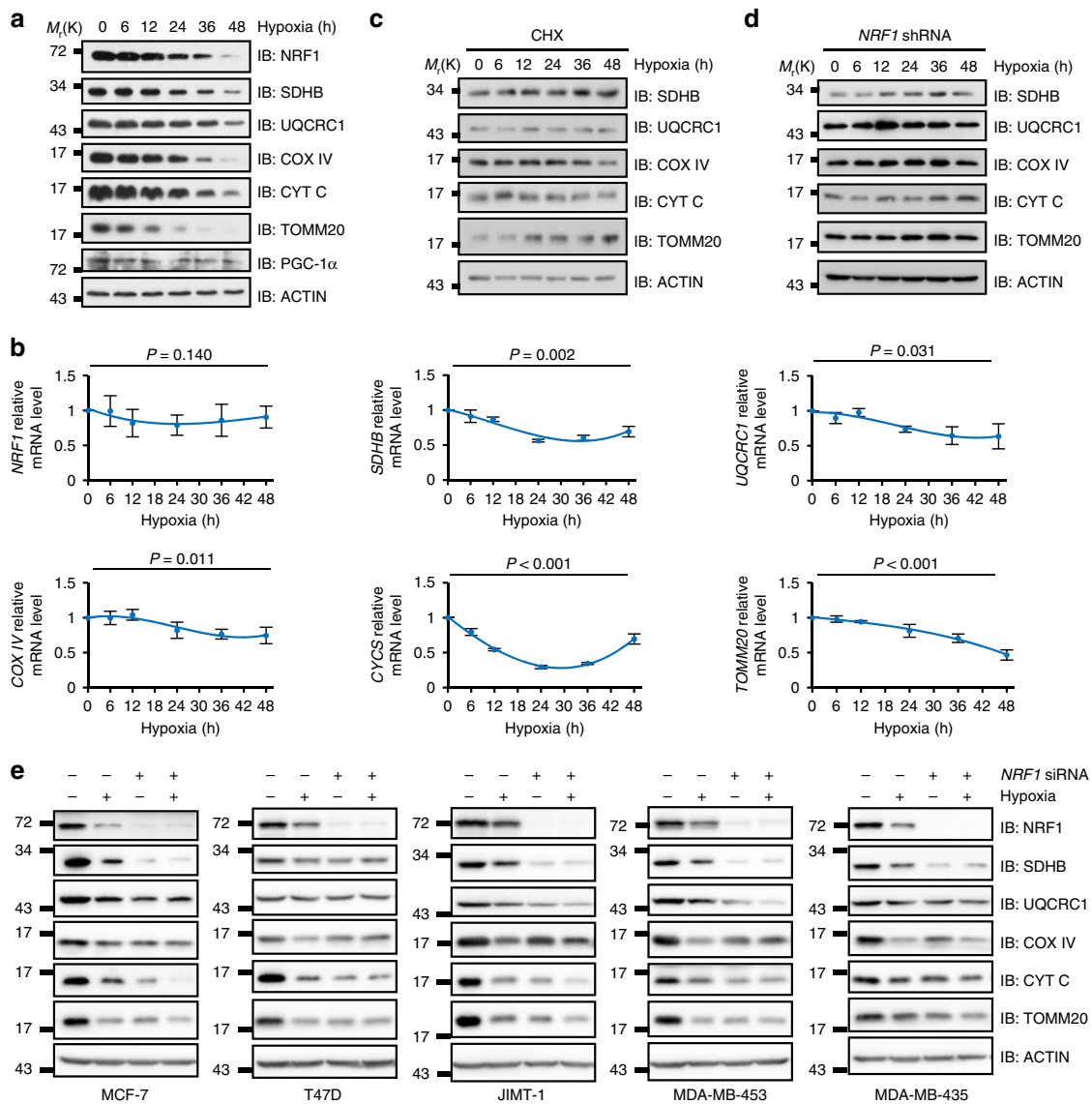

**Fig. 2** Hypoxia transcriptionally regulates nuclear-encoded mitochondrial gene expression through NRF1. **a**, **b** Analysis of protein levels determined by immunoblotting using the indicated antibodies (**a**), and of mRNA levels, determined by real-time PCR (qRT-PCR) for the indicated genes (**b**), at the indicated time points in MDA-MB-231 cells cultured under hypoxia. qRT-PCR results were normalized to the housekeeping gene *B2M*. Detailed statistical data of (**a**) are shown in Supplementary Fig. 1a. **c** MDA-MB-231 cells were pretreated with 25 μg ml$^{-1}$ cycloheximide (CHX) for 2 h under normoxia, then cultured under hypoxic conditions for the indicated time points. Cells were harvested and analyzed by immunoblotting using the indicated antibodies. Detailed statistical data are shown in Supplementary Fig. 2b. **d** Stable *NRF1*-knockdown MDA-MB-231 cells cultured under hypoxia were analyzed by immunoblotting using the indicated antibodies at the indicated time points. Detailed statistical data are shown in Supplementary Fig. 2c. **e** *NRF1* siRNA was transfected into indicated cell lines and cultured under hypoxia for 36 h and then cell lysates were analyzed by immunoblotting using the indicated antibodies. For all panels, error bars indicate s.d., $n = 3$ biological replicates, average of $n = 3$ technical replicates for each biological replicate was used in (**b**). One-way ANOVA was used to compare data

Moreover, we found that knockdown of NRF1 mimicked hypoxia-induced downregulation of mitochondrial markers under normoxic condition (Fig. 5c), which was reversed by re-introduction of NRF1, indicating that the reduction of mitochondrial markers was specifically caused by *NRF1* knockdown instead of off-target effects. We also found that ectopically expressed NRF1 completely rescued hypoxia-induced reduction of mitochondrial markers at both the protein level (Fig. 5c) and the transcriptional level (Fig. 5d), further supported that NRF1 degradation is responsible for the reduction of mitochondrial markers under hypoxia. Furthermore, we examined mitochondria response under hypoxia with the hypoxia-resistant mutant

K230R. We generated a K230R stable cell line from NRF1-knockdown cells and found that mitochondrial markers were significantly elevated in both normoxic and hypoxic conditions (Fig. 5c). Time-course experiments also showed that those mitochondrial markers were no longer responsive to hypoxia in K230R cells (Fig. 5e and Supplementary Fig. 2d).

Importantly, the transcriptional level of SIAH2 was found elevated (Supplementary Fig. 5b) and negatively correlated with the expression of many NEMGs in breast cancer samples (Supplementary Fig. 5c). This provides evidence that SIAH2 may facilitate negative regulation of NEMG expression in cancer. Taken together, we conclude that the SIAH2-NRF1 axis functions

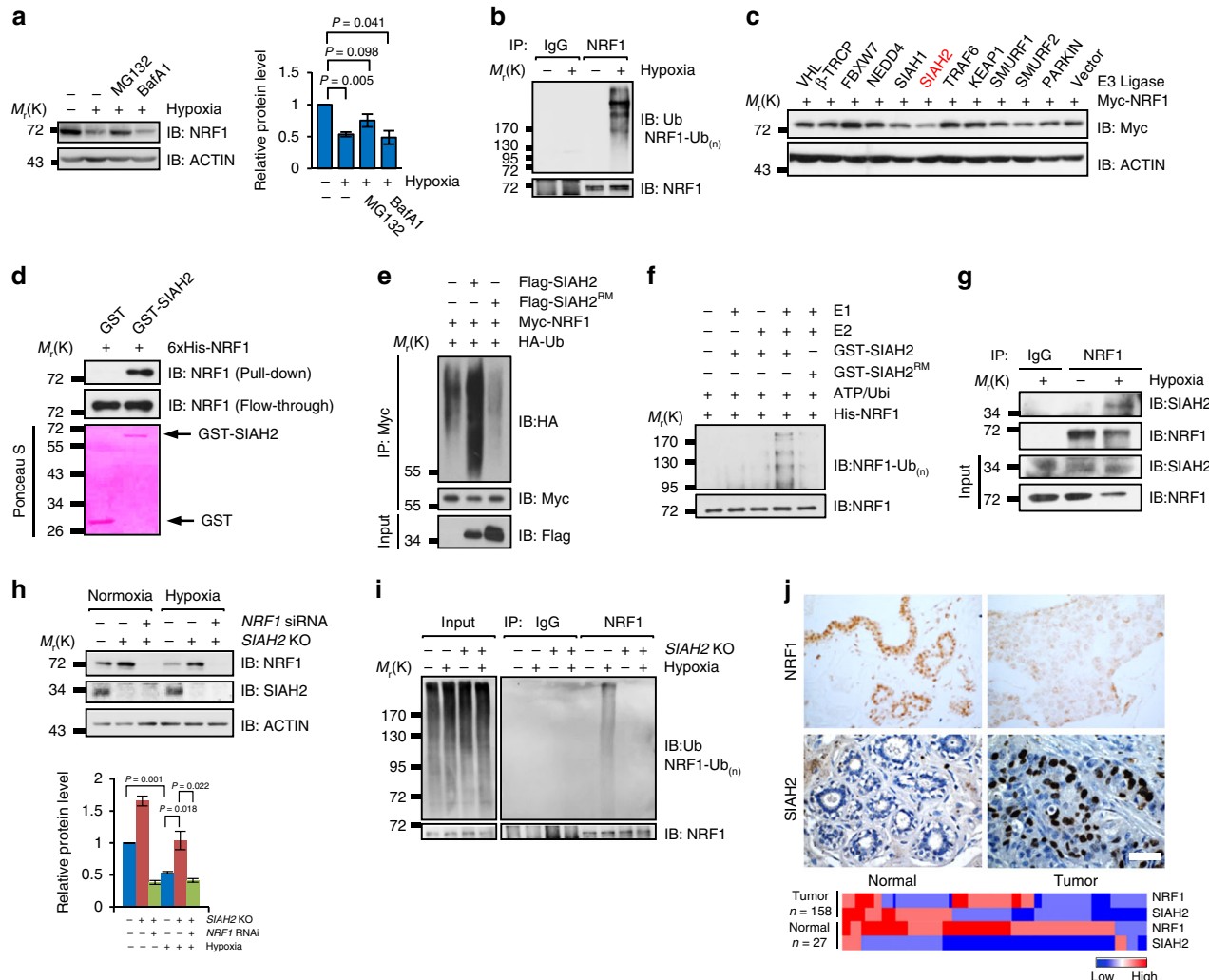

**Fig. 3** SIAH2 promotes NRF1 polyubiquitination and degradation under hypoxia. **a** Hypoxia-mediated NRF1 degradation is inhibited by the proteasomal inhibitor MG132, but not by the autophagy inhibitor Bafilomycin A1 (BafA1). Right: densitometric quantification of NRF1 expressions. **b** MDA-MB-231 cells were incubated under normoxia or hypoxia for 24 h. Cells were treated with 10 μM MG132 for 6 h before harvesting. Cell lysates were immunoprecipitated with anti-NRF1 antibodies and then detected by western blotting with anti-NRF1 and anti-Ubiquitin antibodies. **c** Hypoxia-related ubiquitin E3 ligases were transiently co-transfected with Myc-NRF1 into HeLa cells for 24 h, and Myc-NRF1 protein levels were detected by western blotting with anti-Myc antibodies. **d** Direct interactions between bacterially expressed His-NRF1 and GST-SIAH2 in vitro. **e** Ectopic expression of SIAH2, but not SIAH2[RM] increased NRF1 ubiquitination in vivo. **f** Ubiquitination of bacterially expressed His-NRF1 by purified SIAH2 but not by SIAH2[RM] in vitro. **g** MDA-MB-231 cells were cultured under normoxia or hypoxia for 18 h, then treated with 10 μM MG132 and incubated under normoxia or hypoxia for another 6 h. Endogenous interactions between NRF1 and SIAH2 were analyzed by immunoprecipitation. **h** Wild-type or $SIAH2^{-/-}$ MDA-MB-231 cells were transiently transfected with scramble or NRF1-targeted siRNA, and then cultured under normoxia or hypoxia for 36 h. Cells were harvested and analyzed by western blotting. Bottom: densitometric quantification of the indicated proteins. **i** Hypoxia-induced ubiquitination of NRF1 is abolished by depletion of SIAH2 in vivo. **j** Top: representative immunohistochemical staining of NRF1 and SIAH2 in normal breast tissue and breast cancer tissue from the tissue microarray. (Brown color indicates positive immune reaction; scale bars, 50 μm). Bottom: heat map showing quantitative analysis of the expression of NRF1 and SIAH2 proteins in normal breast tissues and breast cancer tissues. $n = 158$ breast tumors and $n = 27$ normal breast samples. Statistical analysis of the immunostaining results is shown in Supplementary Table 3, 4. For all panels, error bars indicate s.d., $n = 3$ biological replicates. Data were compared with two-tailed paired ratio $t$-tests

as an endogenous regulator of NEMG expression in response to hypoxia.

**The SIAH2-NRF1 axis facilitates metabolic reprogramming.** Next, we asked how hypoxia and SIAH2-mediated mitochondrial alterations would impact cellular metabolism and physiological functions. We found that *SIAH2*-knockdown cells showed significantly elevated succinate dehydrogenase (SDH) activity under both normoxic and hypoxic conditions (Supplementary Fig. 6a). Moreover, ATP levels, NAD$^+$/NADH ratios, oxygen consumption and mitochondrial mass (Supplementary Fig. 6b-f) all

consistently elevated in *SIAH2*-knockdown cells. Fatty acid levels were reduced in SIAH2-knockdown cells when cultured under hypoxia (Supplementary Fig. 6g), which was further validated in *SIAH2*-deficient xenograft tumor tissues by weakened Oil Red staining (Supplementary Fig. 6h). These data were consistent with a previous report showing that *SIAH2*-knockdown cells had a deficiency in fatty acid synthesis[35]. However, palmitic acid treatment, which gives overdose of fatty acid to exclude the influence from newly synthesized fatty acid, of *SIAH2*-knockdown cells caused reduced Oil Red staining intensity regardless of the oxygen levels (Supplementary Fig. 6i), implying that fatty acid

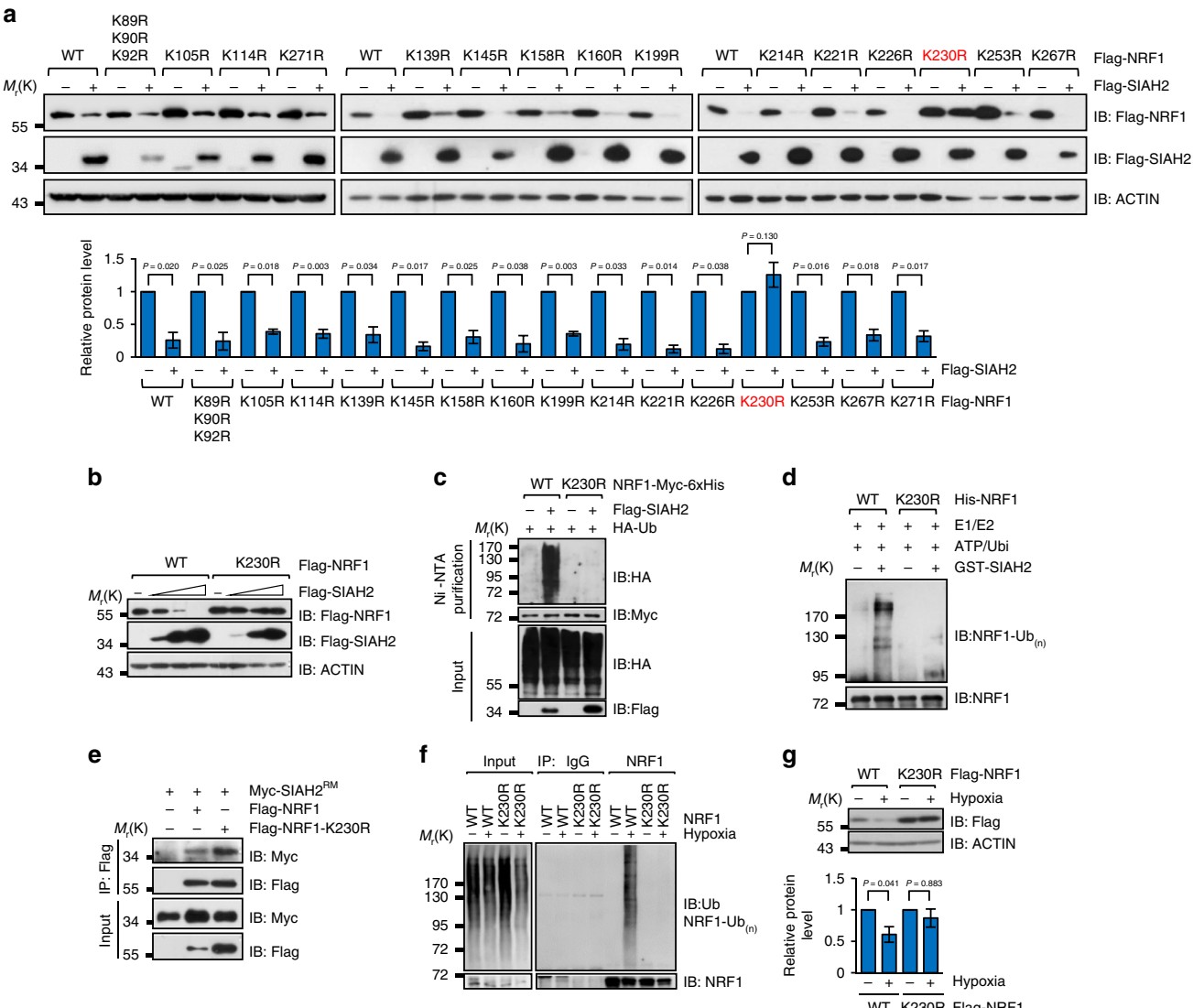

**Fig. 4** NRF1 Lys230 is responsible for SIAH2-mediated NRF1 ubiquitination and degradation under hypoxia. **a** Determination of the Lysine 230 of NRF1 is responsible for SIAH2-induced degradation. Quantification of NRF1 protein levels is shown below. **b** SIAH2 induces degradation of wild-type NRF1 in a dosage-dependent manner but has no effect on NRF1-K230R. **c** SIAH2 induces ubiquitination of NRF1 but not NRF1-K230R. **d** Purified SIAH2 promotes ubiquitination of bacterially expressed wild-type NRF1 but not NRF1-K230R in vitro. **e** Co-immunoprecipitation of exogenously expressed Myc-NRF1 or Myc-NRF1-K230R with Flag-SIAH2[RM]. **f** Hypoxia-induced ubiquitination of NRF1 is abolished by expression of the NRF1-K230R mutant in vivo. **g** HeLa cells were transiently transfected with wild-type NRF1 or NRF1-K230R for 12 h, and then incubated under normoxia or hypoxia for an additional 36 h. Cells were harvested and analyzed by immunoblotting with the indicated antibodies. Quantification of NRF1 protein levels is shown below. For all panels, error bars indicate s.d., $n = 3$ biological replicates. Data were compared with two-tailed paired ratio $t$-tests

consumption may be also increased in *SIAH2*-deficient cells. Moreover, *SIAH2*-knockdown cells also showed decreased glucose consumption when cultured under hypoxic condition (Supplementary Fig. 6j), probably due to increased mitochondrial function[36]. PGE2 and lactate (a glycolysis product) are synthesized from fatty acids[37,38] and glucose[3] respectively. We found that both two factors were downregulated in *SIAH2*-knockdown cells compared with wild-type cells when cultured under hypoxic condition (Supplementary Fig. 6k, l), indicating that SIAH2 is required for hypoxia-induced metabolic reprogramming.

We then checked whether the SIAH2-NRF1 axis was involved in these processes. Consistent with our previous data, *SIAH2*-deficient cells showed significantly reduced PGE2 and lactate production, along with reduced glucose consumption comparing with wild-type cells when cultured under hypoxia (Fig. 6a–c). Simultaneous knockdown of NRF1 completely rescued the PGE2

and lactate levels (Fig. 6a–c). Further analysis showed that the transcription level of *pyruvate dehydrogenase beta* (*PDHB*) (Fig. 6d), encoding an essential component of pyruvate dehydrogenase (PDH) complex which couples glycolysis and Krebs cycle and determines the fate of pyruvate[39], was significantly upregulated in *SIAH2*-deficient cells. However, this effect was abrogated by simultaneous knockdown of NRF1 (Fig. 6d), indicating that *PDHB* is a potential target gene of NRF1. Indeed, this was proved by endogenous chromatin immunoprecipitation (ChIP) assays and a conserved NRF1 binding site was identified in the third intron of the *PDHB* gene (Fig. 6e, f). These data demonstrate that SIAH2-NRF1 axis facilitates hypoxia-induced metabolic reprogramming by limiting mitochondrial function via inhibiting NEMG expression. In addition, stably expressed wild-type NRF1 and the hypoxia-resistant mutant K230R significantly reversed the hypoxia-induced production of

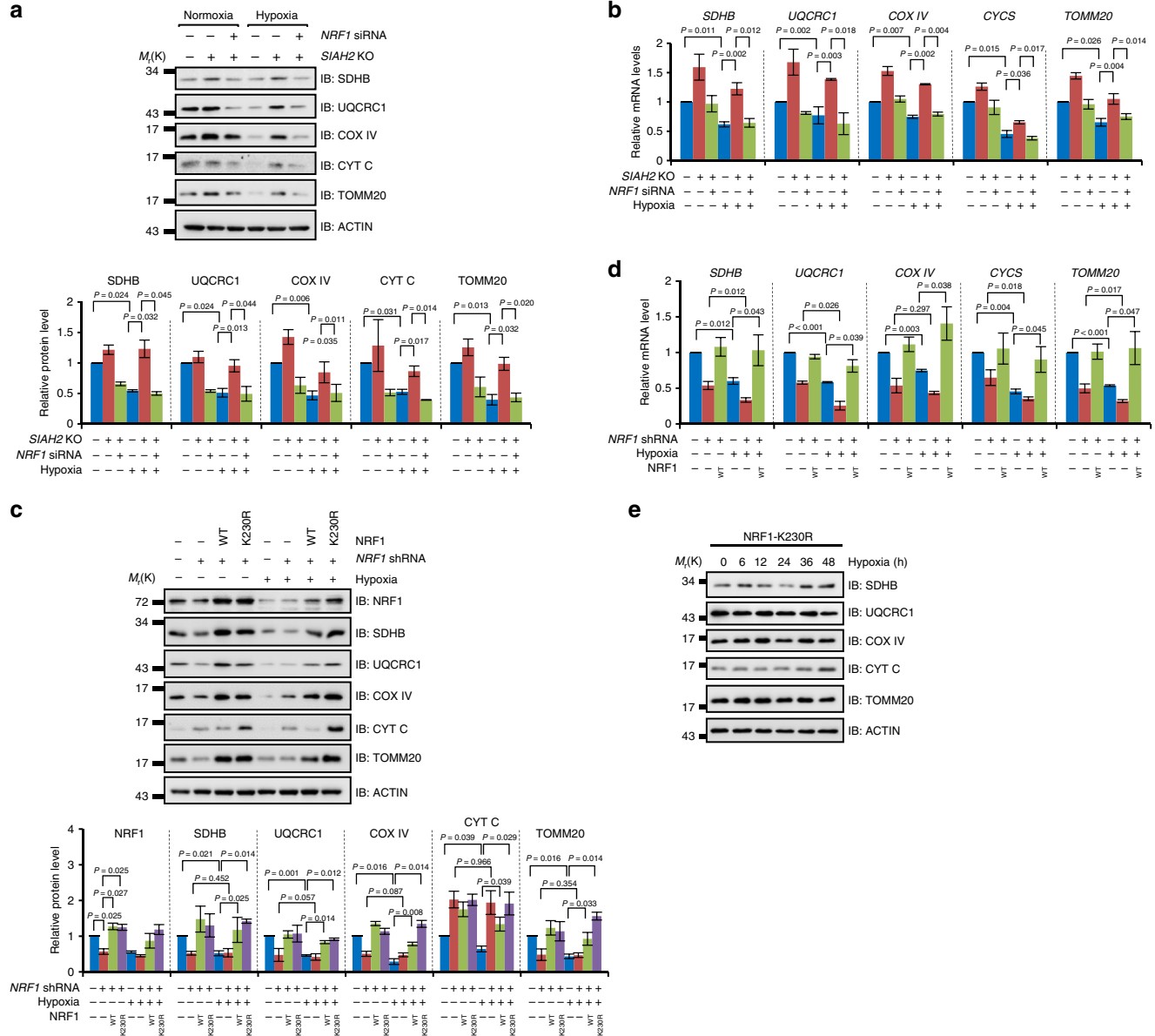

**Fig. 5** SIAH2 regulates nuclear-encoded mitochondrial gene expression through NRF1. **a** Wild-type or *SIAH2*⁻/⁻ MDA-MB-231 cells were transiently transfected with scramble or *NRF1*-targeted siRNA then cultured under normoxia or hypoxia for 36 h. Cells were harvested and analyzed by western blotting. Bottom: densitometric quantification of the indicated proteins. **b** Statistical analysis of qRT-PCR data from cells treated as in (**a**). qRT-PCR results were normalized to the housekeeping gene *B2M*. **c** Stable *NRF1*-knockdown MDA-MB-231 cells reconstituted with wild-type NRF1 or the NRF1-K230R mutant, together with mock and *NRF1*-knockdown MDA-MB-231 cells, were cultured under normoxia or hypoxia for 36 h and analyzed by immunoblotting. Bottom: densitometric quantification of the indicated proteins. **d** Cells treated as in (**c**) were analyzed by qRT-PCR and the data were statistically compared. qRT-PCR results were normalized to the housekeeping gene *B2M*. **e** MDA-MB-231 cells stably expressing K230R-NRF1 were cultured under hypoxia and then analyzed by immunoblotting using the indicated antibodies at the indicated time points. Detailed statistical data are shown in Supplementary Fig. 2d. For all panels, error bars indicate s.d., *n* = 3 biological replicates, average of *n* = 3 technical replicates for each biological replicate was used in (**b**) and (**d**). The two-tailed paired ratio *t*-test was used to compare data in (a-d) and one-way ANOVA was used to compare data in (**e**)

PGE2 and lactate and the consumption of glucose (Fig. 6g-i), along with increased mitochondrial mass (Fig. 6j). These data further support our notion that hypoxia-induced NRF1 degradation through SIAH2 is important for cellular metabolic reprogramming under hypoxia.

It is well known that hypoxia-induced factors (HIFs) transcriptionally regulate the expression of glycolytic genes under hypoxia, favoring a shift from oxidative phosphorylation to glycolysis[40–43]. Thus, we used the hypoxia-resistant mutant K230R to test whether hypoxia-induced NRF1 degradation could affect HIFs-related hypoxia response. No inhibiting effect of

NRF1 degradation was found on hypoxia-induced HIF1/2 stabilization or the activation of their target genes (Fig. 6k). Consistent with *PDHB* as a target gene of NRF1, we found that PDHB was retained in K230R cells regardless of oxygen levels (Fig. 6k) or the PDH activity (Fig. 6l). Thus, hypoxia-induced NRF1 degradation is responsible for *PDHB* downregulation and subsequently facilitates the deactivation of PDH, enhancing the conversion of pyruvate to lactate. Taken together, we conclude that HIFs stabilization and NRF1 degradation regulate the hypoxia-induced metabolic reprogramming process in a cooperative but parallel fashion (Fig. 6m).

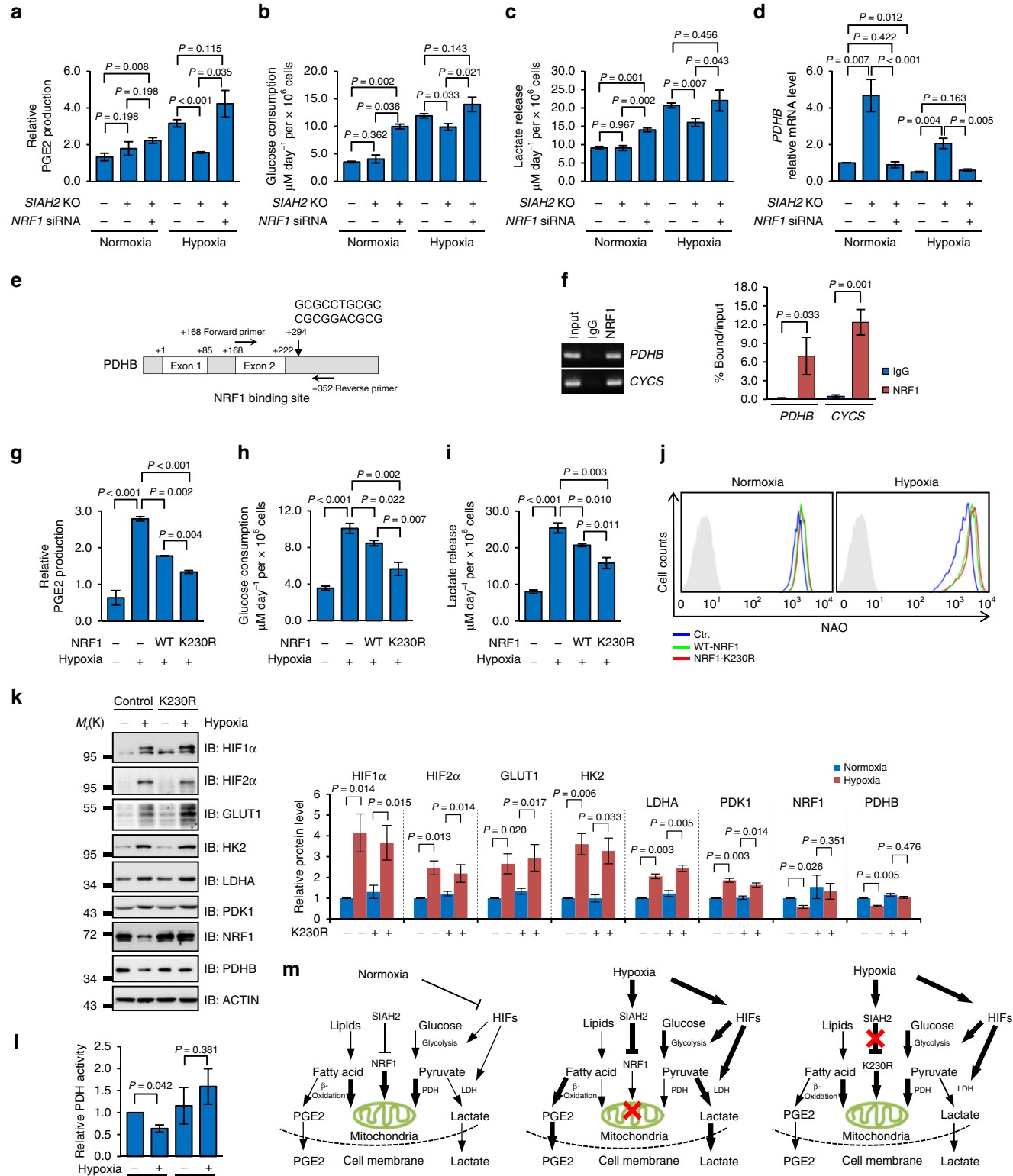

**NRF1 degradation is required for tumor maintenance.** It is noticeable that *SIAH2*-deficient cells showed decreased proliferation and retarded growth in xenograft experiments (Supplementary Fig. 7a-e). However, simultaneous knockdown of NRF1 had no effect on this phenotype (Supplementary Fig. 7a-e), probably due to other signal pathways involved and the necessities of mitochondria in supporting tumor growth[10–13], considering NEMGs were reduced in *NRF1* knockdown cells.

To further address the impact of NRF1 degradation on tumorigenesis, we performed xenograft experiments with wild-type control cells, *NRF1* stable knockdown cells and *NRF1* stable knockdown cells that stably reconstituted with wild-type NRF1 or the hypoxia-resistant mutant K230R. The results showed that *NRF1* knockdown inhibited the tumor growth, whereas reconstituted wild-type NRF1 could completely reverse this growth retardation phenotype (Fig. 7a–c). The intact surgical removed

**Fig. 6** The SIAH2-NRF1 axis facilitates hypoxia-induced metabolic reprogramming. **a–d** Wild-type, *SIAH2*[−/−] and *SIAH2*[−/−]/*NRF1* siRNA MDA-MB-231 cells were cultured under normoxia or hypoxia for 36 h, and concentrations of prostaglandin E2 (PGE2) within cells (**a**), glucose (**b**), lactate (**c**) in the culture medium and mRNA levels of *PDHB* (**d**) were analyzed. qRT-PCR results were normalized to the housekeeping gene *B2M*. **e** Diagram showing the NRF1 binding site in *PDHB* gene and oligonucleotides used in the ChIP assay. **f** ChIP assay was performed with IgG and antibody against NRF1 and indicated genes were analyzed by qRT-PCR. qRT-PCR results were normalized to the input. **g–i** MDA-MB-231 cells, either mock-treated or stably expressing wild-type NRF1 or the NRF1-K230R mutant, were cultured under normoxia or hypoxia for 36 h, and concentrations of prostaglandin E2 (PGE2) (**g**), glucose (**h**) and lactate (**i**) were analyzed. **j** Indicated cells were cultured under normoxia or hypoxia for 36 h, cells were stained with 100 nM Nonyl Acridine Orange (NAO) and analyzed. **k** Wild-type or K230R stably expressed MDA-MB-231 cells were cultured under normoxia or hypoxia for 36 h. Cells were harvested and analyzed by western blotting with indicated antibodies. Right: densitometric quantification of the indicated proteins. **l** Cells from (**k**) were collected and PDH activity was measured. **m** A proposed model of the SIAH2-NRF1 axis in regulating hypoxia-induced metabolic reprogramming. For all panels, error bars indicate s.d., n = 3 biological replicates, average of n = 3 technical replicates for each biological replicate was used in (**a–d**), (**f–i**) and (**l**). The two-tailed unpaired student *t*-test was used in (**a–c**) and (**f–i**). Two-tailed paired ratio *t*-test was used in (**d**) and (**k–l**)

K230R tumors showed almost the same volume and weight comparing with control groups (Fig. 7a–c). However, histological analysis of the xenograft tumor tissues revealed that a slight and a dramatic increase of necrotic areas in wild-type NRF1 and K230R reconstituted tumor tissues respectively (Fig. 7d, e).

Considering tumor progression depends on tumor maintenance in addition to cell proliferation, mitochondria are required for tumor growth and are reduced in the tumors during the process of tumor progression due to hypoxia-induced NRF1 degradation. We hypothesize that hypoxia-induced NRF1 heterogeneous expression may function differently during tumor progression. In normoxic regions, NRF1 induces NEMG expression and contributes to cell proliferation. Whereas in hypoxic regions, NRF1 degradation-mediated inhibition of mitochondrial function via decreased NEMG expression can inhibit cell growth, which is consistent with the growth inhibitory effect of *NRF1* knockdown under normoxia (Supplementary Fig. 7f). However, NRF1 degradation can trigger metabolic reprogramming, such as increased PGE2 and lactate. Both of these metabolites are important immune factors capable of polarizing tumor-associated macrophages (TAMs)[44–46], which are abundant in tumor tissues and have various functions such as promoting metastasis and angiogenesis, inhibiting immune surveillance and promoting tumor maintenance[44–47].

We thus examined whether altered NRF1 would affect TAMs in tumor tissues. Immunofluorescence data showed that polarized TAMs (ARG1 positive cells) were abundant in *NRF1* knockdown tumors and accompanied with reduced mitochondrial marker TOMM20, whereas reconstituted wild-type NRF1 and K230R strikingly reversed these phenotypes (Fig. 7f). This data indicates that NRF1 level impacts the polarization of TAMs. Consistently, in vitro macrophage polarization assay further demonstrated that the mediators in polarizing the macrophages were small molecules and the conditioned medium from K230R cells had significant defect in polarizing bone-marrow derived macrophages (BMDMs) toward the M2 phenotype (Fig. 7g). We also found that in spontaneous tumor tissue, ARG1 positive TAMs were enriched in hypoxic regions (GLUT1 positive regions) and accompanied with reduced mitochondrial marker TIMM23 (Fig. 7h), further indicating that hypoxia-induced reduction of NEMG expression via NRF1 degradation contributes to the polarization of TAMs.

**Accumulated NRF1 enhances apoptosis and impairs efferocytosis.** Even though increased NRF1 may impact the polarization of TAMs, deficiency in TAMs polarization seems not enough to explain the dramatic necrosis in K230R tumor tissues. Reconstituted wild-type NRF1 tumors also showed reduced M2-TAMs in the tumor tissues (Fig. 7f), however, only slightly necrosis was shown (Fig. 7d, e). Hence, firstly we checked whether alterations of NRF1 could induce necroptosis, which is a

form of programmed cell death mediated by MLKL[48,49]. Western blot analysis of the xenograft tumor tissues showed that the ratio of activated MLKL (p-MLKL Ser358) to total MLKL was decreased in reconstituted wild-type NRF1 or K230R cells comparing with control cells (Supplementary Fig. 8), indicating that the necrosis in the tissues is not due to MLKL-mediated necroptosis.

Next, we checked whether accumulated NRF1 could affect the apoptotic process. Strikingly, dramatic accumulation of apoptotic cells (TUNEL positive) was found in K230R tissues (Fig. 8a, b). Under normal condition, apoptotic cells are efficiently eliminated by macrophages to prevent inflammation[50]. If apoptotic cells are not efficiently eliminated by macrophages, those apoptotic cells will rupture and cause a secondary necrosis[51]. In K230R tissues, however, this process was significantly inhibited, seen as dramatically increased free apoptotic cells (ACs) without association with macrophages (Fig. 8c) comparing with wild-type control or *NRF1* knockdown tumors, indicating that the process of efferocytosis in K230R tissues is impaired, which is consistent with the consequence that lacking of polarized M2-macrophages, a major sub-type of macrophages that responsible for the clearance of apoptotic cells[52].

We then asked why more apoptotic cells were shown in K230R tissues. Intriguingly, we found that *FADD*, a gene encodes a core component of DISC during apoptosis[53], has a conserved NRF1 binding site in the beginning of its first exon (Fig. 8d). ChIP assay further demonstrated that *FADD* is a target gene of NRF1 (Fig. 8e). Gene expression analysis showed that *FADD* expression was downregulated in either *NRF1* knockdown cells or under hypoxia, and this downregulation induced by hypoxia was completely abrogated in K230R cells (Fig. 8f), which was consistent with the expression pattern of other NRF1 target genes. Hypoxia-induced FADD downregulation depending on degrading NRF1 was also confirmed at protein level by western blot analysis (Fig. 8g). We also showed that FADD was more abundant in reconstituted NRF1 and K230R xenograft tumor tissues (Fig. 8h), indicating that the increased FADD may be responsible for the dramatic apoptosis in K230R tissues. Since TRAIL (TNF-Related Apoptosis-Inducing Ligand) plays a vital role in immune surveillance against tumors in vivo[54], we mimicked the tumor microenvironment in the context of hypoxia and TRAIL, and tested the impact of NRF1 degradation on cell apoptosis. The results suggested that hypoxia could alleviate TRAIL-induced apoptosis in wild-type cells, which was mimicked by *FADD* knockdown under normoxia (Fig. 8i), validated by the expressions of cleaved PARP1 and activated Caspase-3. TRAIL-treated K230R cells showed significantly enhanced apoptosis not only under normoxia but also under hypoxia comparing with wild-type cells (Fig. 8i). These phenotypes were completely reversed by *FADD* knockdown (Fig. 8i), indicating that the degradation of NRF1 is required for hypoxic inhibition of

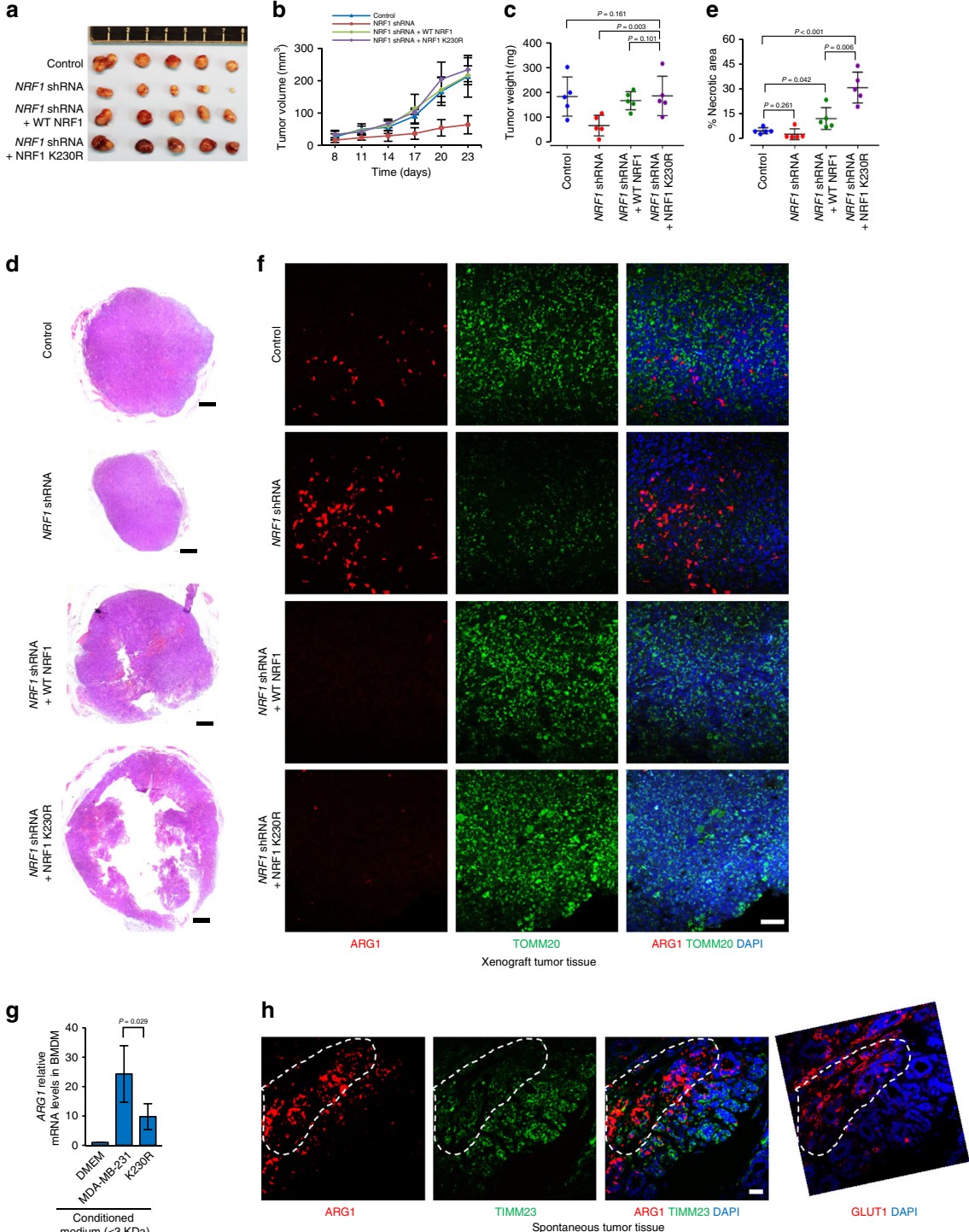

**Fig. 7** NRF1 degradation is required for tumor maintenance and TAM polarization. **a**–**c** Images (**a**), growth curves (**b**) and weights (**c**) of xenograft tumors derived from MDA-MB-231 cells with the indicated modifications. Tumors were established in mice by subcutaneous injection of cells. **d**, **e** The indicated xenograft tumor tissues were analyzed by hematoxylin-eosin staining (**d**) and tissue cross-sections were quantified necrotic area (**e**). Scale bars, 500 μm. **f** The indicated xenograft tumor tissues were stained with anti-ARG1 (red) and anti-TOMM20 (green) antibodies together with DAPI (blue). Scale bars, 50 μm. **g** <3 KDa fractions from DMEM or indicated cell-conditioned medium were used to stimulate bone-marrow derived macrophages. *ARG1* mRNA expression was analyzed by qRT-PCR and normalized to *ATCB* as housekeeping gene. **h** Tissues derived from mouse spontaneous breast cancer were stained with DAPI (blue) together with anti-ARG1 (red) and anti-TIMM23 (green) or anti-GLUT1 (red) antibodies. Scale bars, 25 μm. For all panels, error bars indicate s.d. For panel (**a**–**f**) and (**h**), *n* = 5 mice per group. For panel (**g**), *n* = 3 biological replicates, average of *n* = 3 technical replicates for each biological replicate was used. The two-tailed unpaired student *t*-test was used to compare data

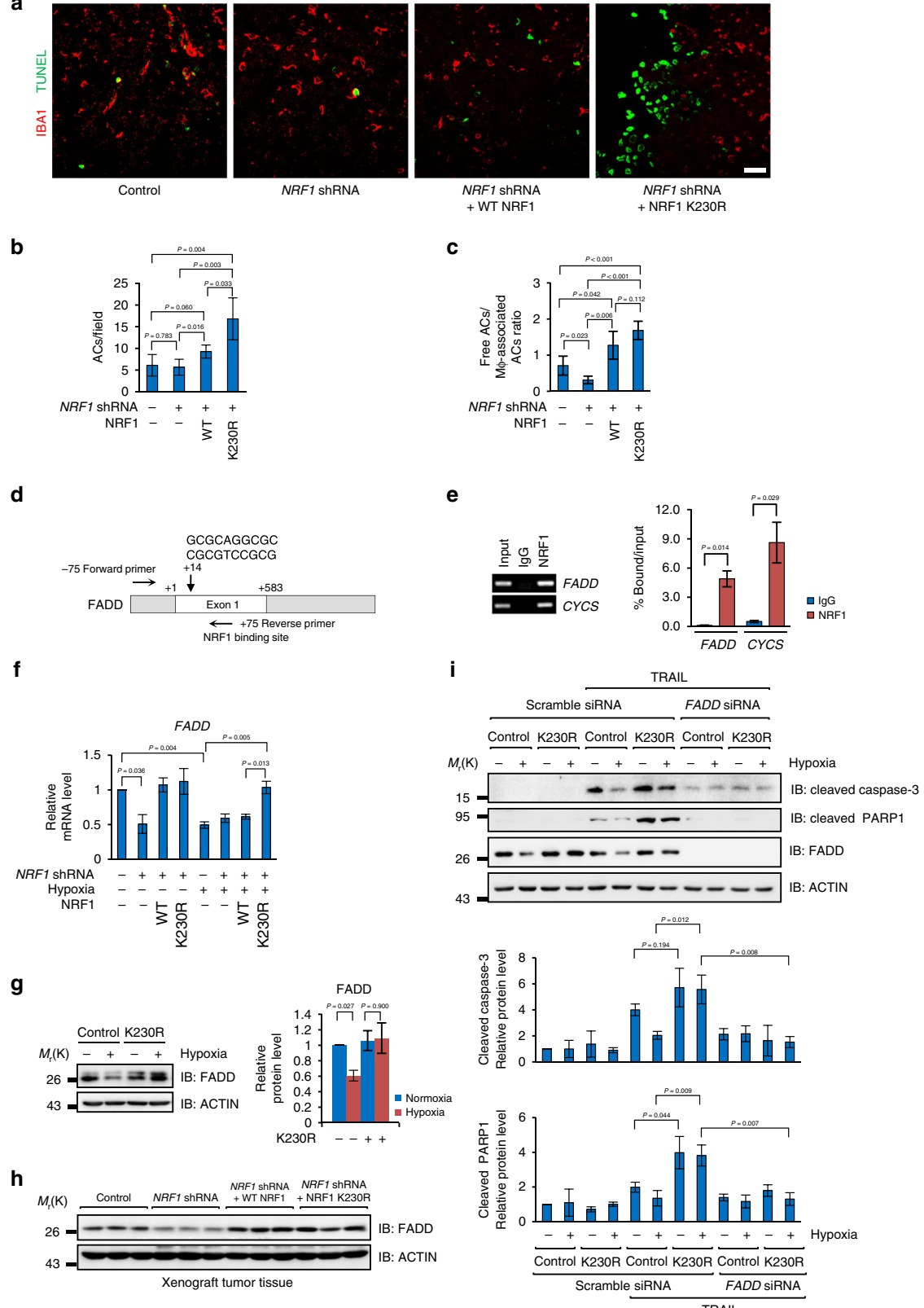

apoptosis and the constant expression of FADD is responsible for the enhanced apoptosis in TRAIL-treated K230R cells.

Taken all these data together, we conclude that the accumulated NRF1 can sustain the expression of FADD under hypoxia, which may enhance the susceptibility of tumor cells in response to extrinsic death stimuli, such as TRAIL, and lead to increased apoptosis. Besides, accumulated NRF1 also inhibits the polarization of TAMs, which leads to defects of TAMs in elimination of apoptotic cells. Hence, these two aspects together cause secondary necrosis and subsequently compromise tumor maintenance.

**Fig. 8** NRF1 accumulation enhances FADD-dependent apoptosis and impairs efferocytosis in vivo. **a–c** The indicated xenograft tumor tissues were stained with anti-IBA1 (red) and TUNEL (green) (**a**) and quantified TUNEL + apoptotic cells (ACs) (**b**), and ratio of free ACs: macrophage-associated ACs (**c**). Scale bars, 25 μm. **d** Diagram showing the NRF1 binding site in *FADD* gene and oligonucleotides used in the ChIP assay. **e** ChIP assay was performed with IgG and antibody against NRF1 and indicated genes were analyzed by qRT-PCR. qRT-PCR results were normalized to the input. **f** The indicated cells were cultured under normoxia or hypoxia for 36 h and *FADD* mRNA levels were statistical analyzed. qRT-PCR results were normalized to the housekeeping gene *B2M*. **g** Wild-type or K230R stably expressed MDA-MB-231 cells were cultured under normoxia or hypoxia for 36 h. Cells were harvested and analyzed by western blotting with anti-FADD and anti-ACTIN antibodies. Right: densitometric quantification of FADD protein levels. **h** Three groups of fresh frozen tissues from indicated xenograft tumors were analyzed by western blotting with the anti-FADD and anti-ACTIN antibodies. **i** Wild-type or K230R stably expressed MDA-MB-231 cells were cultured under normoxia or hypoxia for 24 h and then were treated with 50 ng mL$^{-1}$ TRAIL for additional 12 h. Cells were harvested and analyzed by western blotting with indicated antibodies. Bottom: densitometric quantification of cleaved Caspase-3 and cleaved PARP1 protein levels. For all panels, error bars indicate s.d. For panel (**a–c**), n = 5 mice, average of n = 5–10 pictures per mouse were statistically analyzed. For other panels, n = 3 biological replicates, average of n = 3 technical replicates for each biological replicate was used in (**e–f**). The two-tailed unpaired student t-test was used in (**b–c**) and (**e**). The two-tailed paired ratio t-test was used in (**f–g**) and (**i**)

## Discussion

In this study we have revealed a pathway by which mitochondrial function is spatially reprogrammed in response to microenvironmental cues such that the expression of NEMGs and mitochondrial function are significantly reduced in hypoxic tumors. Such mitochondria-dependent metabolic reprogramming can in turn organize tumor microenvironment by modulating TAMs polarization and efferocytosis to create a pro-tumorigenesis condition. This pathway is triggered by the low oxygen tension, a common feature in malignant solid tumor tissues and subsequent activation of the E3 ligase SIAH2, which leads to the polyubiquitination and proteasomal degradation of NRF1. The degradation of NRF1 is responsible for the reduction of the expression of NEMGs and mitochondrial function in hypoxic tumor cells. Previous studies have shown that HIFs-dependent mitophagy[55] and HIFs-inhibiting mitochondrial biogenesis[56] negatively regulates mitochondrial mass and mitochondrial biogenesis, respectively. However, it appears that inhibition of NRF1 degradation by either depletion of SIAH2 or introducing a NRF1 hypoxia-resistant mutant K230R is sufficient to block the reduction of the expression of NEMGs and mitochondrial mass loss induced by hypoxia, whereas inhibition of NRF1 degradation has no effect on HIFs stabilization or activation, implying that NRF1 degradation has a dominant role in regulating NEMGs expression under hypoxia and that is HIFs-independent.

Previous studies have also demonstrated that HIFs transcriptionally regulate the expression of glycolytic genes under hypoxia, including *GLUT1*, *HK2*, *LDHA* and *PDK1 etc*[40–42], favoring metabolic reprogramming from oxidative phosphorylation to glycolysis. In our work, we showed that the degradation of NRF1 is required for hypoxia-induced metabolic reprogramming. NRF1 degradation results in the downregulation of *PDHB* and many other mitochondrial genes expression, which facilitates the conversion of pyruvate to lactate and synthesis of other metabolites, such as PGE2. These two molecules were recently found having a role in polarizing M2-TAMs[44–46]. Hypoxia-spatially organized mitochondria may generate gradient distributions of these metabolites within tumor tissues, thus resulting in spatially organized M2-TAMs.

TAMs are often prominent immune cells within the tumor microenvironment[57,58]. Normally, they do not become polarized until they received particular microenvironmental signals[59]. The polarized M2-TAMs have been revealed having pro-tumor functions, such as facilitating cancer initiation, promoting angiogenesis, favoring metastasis and immune suppression[60–64]. Thus, they are recognized as prognostic markers of cancers, as well as potential therapeutic targets[65].

Our results indicate that mitochondrial heterogeneity contributes to the generation of tumor metabolic heterogeneity.

Hence, we suggest that hypoxic suppression of mitochondrial function through SIAH2-NRF1 axis serves as a metabolic switch for the spatial organization of pro-tumor microenvironments. It is possible that during tumor development, the abundance of mitochondria within normoxic regions contributes to tumor growth and expansion, whereas the reduced mitochondrial function within hypoxic regions contributes to activation of pro-tumor immune responses and tumor maintenance, thereby supporting tumor progression (Fig. 9).

We also identified that *FADD*, which encodes a core component of DISC during apoptosis, is an NRF1 target gene and is reduced under hypoxia in an NRF1-dependent manner. This further explains why tumor cells become less sensitive to apoptotic stimuli under hypoxic conditions. Moreover, inhibition of hypoxia-induced NRF1 degradation could dramatically sustain FADD expression level and sensitize the cells to pro-apoptotic signals under hypoxia. The expression level of NRF1 seems not only to affect TAMs polarization, but also to regulate apoptosis. Since M2 macrophages are believed to play a major role in the process of efferocytosis[52], dysregulation of NRF1 will cause defects of tissue homeostasis due to the dysregulated apoptosis and efferocytosis. Hence, we believe that regulation of NRF1 expression via SIAH2 by oxygen tension is a switch for a series of intrinsic and extrinsic cellular reactions during tumor progression, which may be of great importance for tumor maintenance. Our findings reconcile the paradoxical role of mitochondria in tumor formation and also imply that interventions blocking hypoxic suppression of NRF1 may contribute to therapeutic strategies in curing cancer by reactivating immunity and sensitizing the tumor cells to apoptotic stimuli.

## Methods

**Cell culture and transfection**. HeLa, HEK293T, MEF, MCF-7, T47D, JIMT-1, MDA-MB-453, MDA-MB-435, and MDA-MB-231 cell lines were from the American Type Culture Collection and were cultured under conditions specified by the supplier. The cell lines were tested as free from mycoplasma contamination. A gas mixture containing 1% O$_2$, 5% CO$_2$, and 94% N$_2$ was flushed into a hypoxic chamber (Billups-Rothenberg) to achieve hypoxic culture conditions. Plasmids were transfected into cells with polyethylenimine according to the manufacturer's protocols.

**Expression constructs**. The mammalian expression plasmids for NRF1, FBXW7, NEDD4, TRAF6, KEAP1, SMURF1, SMURF2, SIAH1, SIAH2, VHL and PARKIN were generated by PCR amplification of the corresponding cDNA followed by cloning into pFLAG-CMV-4 or pcDNA4-TO-Myc-His-B expression vectors. NRF1 cDNA was cloned into the retroviral vector pLHCX for retroviral expression. The mammalian expression plasmid for β-TRCP was generated by insertion of β-TRCP cDNA in-frame into the pcDNA3.0 vector. For recombinant GST fusion proteins, SIAH2 and NRF1 were cloned into pGEX-4T-1 vector and pET28a vector, respectively. E3 ligase activity dead mutant SIAH2$^{RM}$ (H98A/C101A) and all NRF1 mutants were made using the QuickChange Site Mutagenesis Kit (TransGen Biotech). All of the constructs generated were confirmed by DNA sequencing.

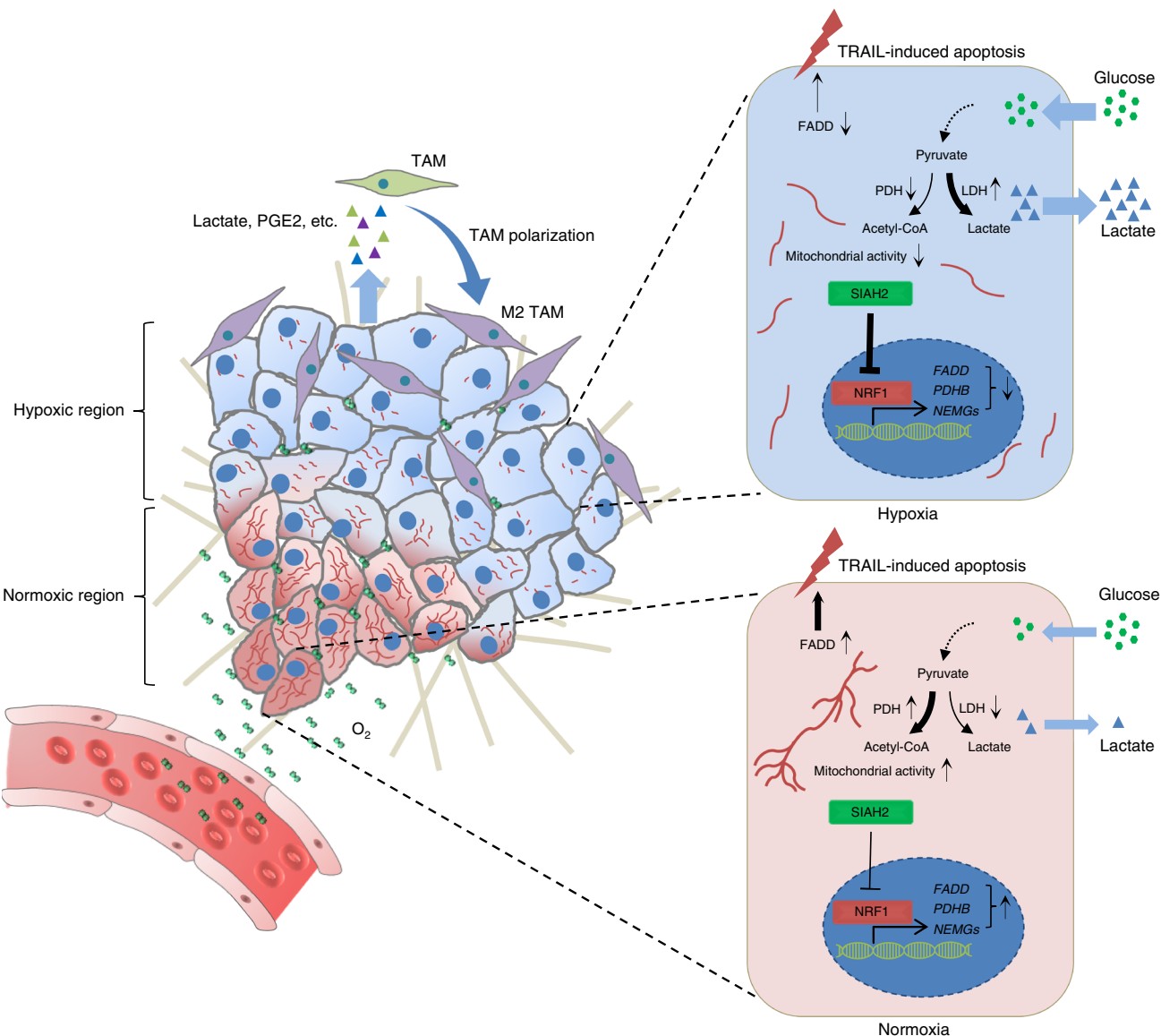

**Fig. 9** A proposed model of the SIAH2-NRF1 axis in regulating the formation of pro-tumor microenvironments

**Immunoblotting**. Cells were lysed in lysis buffer (150 mM NaCl, 20 mM Tris, pH 7.4, 1 mM EGTA, 1 mM EDTA, 1% SDS, 2.5 mM sodium pyrophosphate, 1 mM $Na_3VO_4$, and protease inhibitors (Roche)). Equivalent protein quantities were subjected to SDS-PAGE and transferred to nitrocellulose membranes, then blocked with 5% non-fat milk for 1 h at room temperature. Membranes were then probed with the indicated primary antibodies, followed by the appropriate HRP-conjugated anti-mouse/rabbit (KPL) secondary antibodies. Immunoreactive bands were visualized with chemiluminescence kits (Engreen Biosystem). The following antibodies were used: SIAH2 (1:1000, Proteintech, 12651-1-AP), NRF1 (1:1000, GeneTex, GTX103179), SDHB (1:1000, Abcam, ab14714, clone 21A11AE7), UQCRC1 (1:1000, Proteintech, 21705-1-AP), COX IV (1:1000, Proteintech, 11242-1-AP), Cytochrome C (1:1000, BD Biosciences, 556433, clone 7H8.2C12), TOMM20 (1:1000, BD Biosciences, 612278, clone 29/Tom20), PGC-1α (1:1000, Proteintech, 20658-1-AP), LC3 (1:1000, MBL, PM036), HIF1α (1:1000, Abcam, ab51608, clone EP1215Y), HIF2α (1:1000, Cell Signaling Technology, 7096, clone D9E3), GLUT1 (1:1000, Proteintech, 21829-1-AP), HK2 (1:1000, Cell Signaling Technology, 2106), LDHA (1:1000, Proteintech, 19987-1-AP), PDK1 (1:1000, Proteintech, 18262-1-AP), PDHB (1:1000, Proteintech, 14744-1-AP), FADD (1:1000, Cell Signaling Technology, 2782), p-MLKL (Ser 358) (1:1000, Cell Signaling Technology, 91689, clone D6H3V), MLKL (1:1000, Cell Signaling Technology, 14993, clone D2I6N), PARP (1:2000, Cell Signaling Technology, 9542), Cleaved Caspase-3 (1:1000, Cell Signaling Technology, 9664, clone 5A1E), ACTIN (1:10000, Sigma, A5441, clone AC-15), Flag (1:1,000, Sigma, F1804, clone M2), Myc (1:1,000, Santa Cruz, sc-40, clone 9E10), HA (1:1,000, Abmart, M20003, clone 26D11), His (1:1,000, Santa Cruz, sc-8036, clone H-3) and ubiquitin (1:1,000, Enzo Life Sciences, PW8805, clone FK1). The ImageJ program (http://rsbweb.nih.gov/ij/download.html) was used

for densitometric analyses of western blots, and the quantified results were normalized to the loading control. Uncropped scans of the key blots were shown in Supplementary Fig. 9.

**Immunoprecipitation**. Cells were collected and lysed in 0.5 ml IP lysis buffer (150 mM NaCl, 20 mM Tris, pH 7.4, 1 mM EGTA, 0.5% NP-40, 2.5 mM sodium pyrophosphate, 1 mM $Na_3VO_4$, and protease inhibitors (Roche)) for 1 h on a rotor at 4 °C. After centrifugation at $12,000 \times g$ for 15 min, the cell lysates were immunoprecipitated with 2 µg specific antibodies overnight at 4 °C, then 30 µl protein A/G agarose beads (BioTool, BT-0001-5) were added for an additional 3 h at 4 °C. Thereafter, the precipitants were washed five times with IP lysis buffer, and the immune complexes were boiled with loading buffer for 5 min and analyzed by SDS-PAGE. The following antibodies were used for immunoprecipitations: NRF1 (GeneTex, GTX103179), Myc (Santa Cruz, sc-40, clone 9E10) and Flag (Sigma, F1804, clone M2).

**Immunofluorescence**. Cells were fixed in 4% paraformaldehyde (Dingguo Changsheng Biotechnology) for 15 min at room temperature, then permeabilized with 0.1% Triton X-100 together with DAPI at 4 °C. After blocking in goat serum for 1 h, cells were incubated with primary antibody at 4 °C overnight, then washed three times with PBS and incubated with FITC- or CY3-conjugated secondary antibodies (Invitrogen, 1:1000) for 1 h at room temperature. The slides were then washed three times with PBS and mounted. Cell images were captured with a confocal microscope (Leica). The following antibodies were used for immunofluorescence: GLUT1 (1:200, Abcam, ab15309), TIMM23 (1:500, BD Biosciences,

611223, clone 32/Tim23), TOMM20 (1:200, BD Biosciences, 612278, clone 29/Tom20), LC3 (1:500, MBL, PM036), ARG1 (1:200, Proteintech, 16001-1-AP), and IBA1 (1:100, Proteintech, 10904-1-AP). Fluorescein based TUNEL assay was performed by using an in situ cell death detection kit (ROCHE) according to the manufacturer's protocols.

**Tissue microarrays and immunohistochemistry.** The breast cancer tissue microarrays were purchased from US Biomax. These tissue microarrays consist of 158 analyzable cases of invasive breast carcinoma and 27 analyzable cases of normal breast tissue. For antigen retrieval, the slides were rehydrated and then treated with 10 mM sodium citrate buffer (pH 6.0) heated for 3 min under pressure. The samples were treated with 3% $H_2O_2$ for 15 min to block endogenous peroxidase activity and then blocked with 5% goat serum for 1 h at room temperature. Then the tissues were incubated with the indicated antibodies at 4 ℃ overnight, followed by incubation with HRP-conjugated secondary antibody for 1 h at room temperature. Immunoreactive signal was visualized with a DAB Substrate Kit (MaiXin Bio). Protein expression levels in all the samples were scored on a scale of four grades (negative,+, ++, +++) according to the percentage of immunopositive cells and immunostaining intensity. The $\chi^2$ test was used for analysis of statistical significance. The following antibodies were used for immunohistochemistry: Prohibitin (1:200, Abcam, ab75766, clone EP2803Y), SIAH2 (1:40, Novus Biologicals, NB110-88113, clone 24E6H3) and NRF1 (1:100, Novus Biologicals, NBP1-89125).

**In vivo ubiquitination assay.** Cells were transiently transfected with plasmids expressing HA-ubiquitin and Myc-NRF1 together with Flag-SIAH2 and Flag-SIAH2$^{RM}$. Twenty-four hours after transfection, cells were treated with 10 μM MG132 (Selleckchem, S2619) for 6 h before collection. Cells were washed with cold PBS and then lysed in 200 μl of denaturing buffer (150 mM Tris-HCl, pH 7.4, 1% SDS) by sonication and boiling for 15 min. Lysates were made up to 1 ml with regular lysis buffer and immunoprecipitated with 2 μg anti-c-Myc antibody at 4 ℃ overnight, washed three times with cold lysis buffer and then analyzed by SDS-PAGE. For the NRF1 endogenous ubiquitination assay, cells were treated with 10 μM MG132 for 6 h before collection. Lysates were immunoprecipitated using 2 μg anti-NRF1 antibody and subjected to ubiquitination analysis by western blotting with anti-NRF1 or anti-ubiquitin antibody.

**In vitro ubiquitination assay.** In vitro ubiquitination assays were carried out in ubiquitination buffer (50 mM Tris, pH 7.4, 5 mM $MgCl_2$, 2 mM dithiothreitol) with human recombinant E1 (100 ng, Abcam), human recombinant E2 UbcH5c (200 ng, Upstate), His-tagged ubiquitin (10 μg, Upstate). GST-SIAH2, GST-SIAH2$^{RM}$, His-NRF1 and His-NRF1-K230R were expressed and purified from *Escherichia coli* BL21 cells. Two micrograms of GST, GST-SIAH2 or GST-SIAH2$^{RM}$ proteins was used in the corresponding ubiquitination reactions. Reactions (total volume 30 μl) were incubated at 30 ℃ for 2 h and subjected to ubiquitination analysis by western blotting using anti-NRF1 antibody.

**Lentiviral shRNA cloning, production, and infection.** To generate *SIAH2-* and *NRF1*-knockdown cells, oligonucleotides were cloned into pLKO.1 between the AgeI and EcoRI sites. Lentiviral packaging plasmids psPAX2 and pMD2.G were co-transfected with the backbone plasmid into HEK293T cells for lentivirus production. Cells were selected by 2.5 μg ml$^{-1}$ puromycin in culture medium. The oligonucleotide pair used was as follows: SIAH2 (5′-CCGGGCTGG CTAATAGACA CTGAATCTCGAGATTCAGTGTCTATTAGCCAGCTTTTTG-3′ and 5′-AATTC AAAAAGCTGGCTAATAGACACTGAATCTCGAGATTCAGTGTCTATTAGC CAGC-3′); NRF1 (5′-CCGGGCCACAGCCACACATAGTATACTCGAGTATA CTATGTGTGGCTGTGGCTTTTTG-3′ and 5′-AATTCAAAAAGCCACAGCC ACACATAGTATACTCGAG TATACTATGTGTGGCTGTGGC-3′).

**Retroviral cloning, production, and infection.** To generate MDA-MB-231 cell lines stably expressing wild-type NRF1 and NRF1-K230R, wild-type NRF1 and NRF1-K230R were cloned into the plasmid pLHCX and the constructs were transfected into pAmpho-HEK293T cells for retrovirus production. Supernatants were collected at 48 h after transfection, centrifuged and added to *NRF1* stable knockdown MDA-MB-231 cells for infection. Positive cells were selected by 500 μg ml$^{-1}$ hygromycin in culture medium.

**CRISPR/Cas9-mediated gene-knockout.** To generate *SIAH2* knockout cells, oligonucleotides were cloned into LentiCRISPR. The backbone plasmid was co-transfected with lentiviral packaging plasmids psPAX2 and pMD2.G for lentivirus production. The oligonucleotide pair used was as follows: (5′-CACCGTGCGGG CCCCGGCTCGTCCG-3′ and 5′- AAACCGGACGAGCCGGGGCCCGCAC-3′). Single clones were picked and expanded after 2.5 μg ml$^{-1}$ puromycin selection. Genomic DNA from each line was isolated and the edited genomic regions were amplified by PCR with primers as follows: (5′-ATGAGCCGCCCGTCCTCCAC-3′ and 5′-GGAAACAGGACTGCCGAGGC-3′). The presence of frameshifting indels was confirmed by sequencing. Gene knockout was further confirmed by western blot.

**Knockdown by small interfering RNA (siRNA).** Cells were transiently transfected with siRNAs using Lipofectamine RNAiMAX Transfection Reagent (Thermo-Fisher) with final concentrations at 50 nM (negative control siRNA, RiboBio siN05815122147; Hs_NRF1 siRNA, RiboBio siG14317165344; Hs_FADD siRNA, RiboBio stB0003899C) following the RNAiMAX Transfection Reagent protocols.

**RNA isolation and real-time PCR.** Total RNA was isolated from cultured cells using a total RNA extraction kit (Promega). cDNA was synthesized by reverse transcription using oligo(dT) and subjected to real-time PCR with human NRF1, SDHB, UQCRC1, COX IV, CYCS, TOMM20, PDHB and B2M primers or mouse ARG1 and ACTB primers in the presence of Cyber green PCR-Mix (TransGen Biotech). Relative mRNA abundance was calculated by normalization to B2M mRNA. The following primer pairs were used to detect the mRNA levels of the following genes by qRT-PCR: NRF1 (5′-GCTGATGAAGACTCGCCTTCT-3′ and 5′-TACATGAGGCCGTTTCCGTTT-3′); SDHB (5′-ACAGCTCCCCGTATCAAG AAA-3′ and 5′-GCATGATCTTCGGAAGGTCAA -3′); UQCRC1 (5′-GGGGCAC AAGTGCTATTGC-3′ and 5′-GTTGTCCAGCAGGCTAACC-3′); COX IV (5′-GA GAAAGTCGAGTTGTATCGCA-3′ and 5′-GCTTCTGCCACATGATAACGA-3′); CYCS (5′-CTTTGGGCGGAAGACAGGTC-3′ and 5′-TTATTGGCGGCTG TGTAAGAG-3′); TOMM20 (5′-GGTACTGCATCTACTTCGACCG-3′ and 5′-TG GTCTACGCCCTTCTCATATTC-3′); PDHB (5′-AAGAGGGCCTTTCACTGGA C-3′ and 5′-ACTAACCTTGTATGCCCCATCA-3′); FADD (5′-GCTGGCTCGTC AGCTCAAA-3′ and 5′-ACTGTTGCGTTCTCCTTCTCT-3′); B2M (5′-GAGGC TATCCAGCGTACTCCA-3′ and 5′-CGGCAGGCATACTCATCTTTT-3′); Mouse ARG1 (5′-CTCCAAGCCAAAGTCCTTAGAG-3′ and 5′-AGGAGCTGTC ATTAGGGACATC-3′); Mouse ACTB (5′- GATCTGGCACCACACCTTCT-3′ and 5′- GGGGTGTTGAAGGTCTCAAA-3′). Data were analyzed from three independent experiments and are shown as the average mean ± s.d.

**ChIP assay.** Cells were treated with 1% formaldehyde for 10 min at room temperature then quenched by glycine addition. Cells were harvested by scraping and then sonicated cell lysates were pre-cleared with protein A/G-agarose beads and immunoprecipitated with anti-NRF1 antibody or nonspecific IgG antibody. Agarose beads were extensively washed, eluted with a buffer consisting of 1% SDS. DNA-protein complex were decrosslinked by addition of 10 μg ml$^{-1}$ proteinase K and incubated at 65 ℃ overnight. DNA was obtained by DNA Extraction Kit (Axygen). The following primer pairs were used to detect the enrichment of following genes by qRT-PCR: PDHB (5′-GTCTCCGGGCTGCTGAAGAGG-3′ and 5′-AGGATGCTGGCTCCGCAAACCCAA-3′); FADD (5′-GCCCTCACCGCAGA GAGCTG-3′ and 5′-GCCGCAGCCGTTTCCGCCCT-3′); CYCS (5′-CCGTACAC CCTAACATGCTC-3′ and 5′-TGGCACAACGAACACTCC-3′).

**Xenograft tumorigenesis study.** All mouse experiments were approved by the Institutional Animal Care and Use Committee at the College of Life Sciences at Nankai University. MDA-MB-231 breast cancer cells ($2 \times 10^6$ in 100 μl PBS) were injected subcutaneously into the armpit of six- to eight-week-old female BALB/c nude mice. Tumor size was measured every 3–5 days one week after the implantation and tumor volume was analyzed by using the formula $V = 0.5 \times L \times W^2$ (V: volume, L: length, W: width). The mice were sacrificed and the subcutaneous tumors were surgically removed, weighed and photographed. No statistical method was used to predetermine sample size for each group. The experiments were not randomized.

**Spontaneous mammary tumor induction.** Six-week-old female mice (C57BL/6) were subcutaneously implanted with slow-releasing MPA pellets (50 mg) under anesthesia with Ketaine-xylazine. Seven days after MPA implantation, DMBA in cottonseed oil (200 μL, 5 mg mL$^{-1}$) was orally administered to mice. DMBA administration was repeated at week 2, 3, 5, 6 and 7 after MPA implantation. Twenty weeks after MPA implantation, tumors were detected by manual palpation. The mice were sacrificed and the spontaneous tumors were surgically removed.

**Isolation and culture of bone-marrow derived macrophages.** Mice were euthanized and immersed with 70% ethanol. Femurs and tibiae from C57BL/6 mice were harvested under sterile conditions from both legs, then were immersed with 70% ethanol for 1 min, followed by washing with cold PBS for three times. The bones then were cut and flushed with cold PBS containing 3% heat inactivated fetal bovine serum (FBS) using a 25-gauge needle. The marrow was passed through a 70 μm strainer then centrifuged at $500 \times g$ for 5 min and supernatants were collected and treated with ACK lysing buffer for 2 min. DMEM was then added, and contents were centrifuged at $500 \times g$ for 5 min. The cells were suspended with DMEM containing 10% heat inactivated FBS and 10% L929-conditioned medium then plated in 6-well plate at $2 \times 10^6$ per well and cultured for 7 days to induce macrophage differentiation. The old culture medium was replaced every other day. For the preparation of L929-conditioned medium, $4.7 \times 10^5$ L929 cells were cultured in a T75 flask with 55 mL DMEM containing 10% FBS for 7 days. The supernatant was collected and filtered through a 0.22 μm filter.

**Analysis of metabolites**. Cellular ATP concentrations were determined with an ATP Detection Assay Kit (Beyotime) according to the manufacturer's instructions (catalogue number S0026). Glucose concentrations were analyzed using a blood glucose monitor (Roche). The activity of PDH was determined by micro PDH assay Kit from Solarbio, catalogue number BC0385. To determine the concentrations of other metabolites, kits were purchased from Nanjing Jiancheng Bioengineering Research Institute as follows and used according to the manufacturer's instructions: NAD$^+$/NADH, catalogue number A114; free fatty acids, catalogue number A042-2; PGE2, catalogue number H099; and lactate, catalogue number A019-2. The absorbance was measured by using a Multiskan™ FC Microplate Photometer (ThermoFisher) immediately after sample preparation. Background absorbance was subtracted.

**Mitochondrial SDH activity analysis**. Three thousand cells were seeded into a 96-well plate. Twenty-four hours later, cells were cultured under normoxic or hypoxic conditions for 24 h, then treated with 3-(4, 5-dimethythiazol-2-yl)-2, 5-diphenyl tetrazolium bromide (0.5 mg ml$^{-1}$) for another 5 h. The samples were dissolved in dimethyl sulfoxide and the absorbance was read at 570 nm. Background absorbance at 630 nm was subtracted.

**Oxygen consumption analysis**. Dissolved oxygen was detected by a micro dissolved-$O_2$ electrode (DO-166MT-1, LAZAR Research Laboratories, Los Angeles, CA). Cells were cultured with 2 ml culture medium under normal conditions. The microelectrode was placed at the bottom of the culture dish and a 10-min recording was started immediately after replenishing the culture with fresh medium equilibrated with air. Cell numbers were determined and the oxygen consumption rate was calculated as the change in oxygen concentration/cell number/10 min.

**Bioinformatics analysis**. Datasets GSE15852, GSE18494, GSE25055 and GSE61839 from the Gene Expression Omnibus (GEO) database were analyzed for Gene Ontology (GO) term enrichment using the Gene Set Enrichment Analysis (GSEA) algorithm and DAVID analysis. Oncomine™ (Compendia Bioscience, Ann Arbor, MI) was used for analysis and visualization of the transcriptional levels of NRF1 and SIAH2 in normal and breast tumor samples. cBioPortal was used for the correlation analysis of SIAH2 and NEMGs in clinical breast tumor samples based on the Cancer Genome Atlas (TCGA) database.

**Statistics and repeatability of experiments**. All error bars indicate s.d. Statistical comparisons were made using the two-tailed paired ratio t-test, two-tailed unpaired Student's t-test or one-way ANOVA. The data analyzed by the two-tailed unpaired Student's t-test are normally distributed. For correlations between Prohibitin, SIAH2 and NRF1 protein levels in clinical samples, statistical significance was determined using the $\chi^2$ test. For statistical tests, $p < 0.05$ was used as the criterion for statistical significance. The variance was similar between groups that were being statistically compared and no samples were excluded from the analysis. The experiments were repeated at least three times. The investigators were not blinded to allocation during experiments and outcome assessment.

## Data availability

The gene transcription profiling data referenced during the study are available in public repositories from the Gene Expression Omnibus (GEO) database (https://www.ncbi.nlm.nih.gov/geo/), Oncomine™ (https://www.oncomine.org/) and cBioPortal (http://www.cbioportal.org/). The data from the Gene Expression Omnibus (GEO) database analyzed for this study are GSE15852, GSE18494, GSE25055 and GSE61839. All the other data supporting the findings of this study are available within the article and its Supplementary Information files. All other relevant data are available from the corresponding author upon reasonable request.

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

## Acknowledgements

The authors thank Dr. Jintang Dong from Nankai University for providing the mouse spontaneous breast cancer samples and Dr. Jie Li for proofreading of this manuscript. This work was supported by the National Natural Sciences Foundation of China (91754114, 31701235 and 91849201), Key project of Frontier Science of Chinese Academy of Sciences (QYZDJSSW-SMC004), Fund for Strategic Pilot Technology Chinese Academy of Sciences (XDPB1002), 111 Project from the Ministry of Education and the State Administration of Foreign Experts Affairs of the People's Republic of China (B08011) and the China Postdoctoral Science Foundation (2017M610159).

## Author contributions

B.M. found the initial phenomenon, together with H.C. conceived and designed the experiments with the help from Y.Z. and Q.C. H.C. performed most of the biochemical experiments and transcriptional analysis with the help from T.Z., Q.L., R.C., K. M., and B.M. B.M. performed all bioinformatics analysis and statistical analysis with the help from H.C. B.M., H.C. G.G., and C.M. performed xenograft implantation experiments. B.M. and H.C. performed studies on tissue microarrays of human patient samples. Q.L., R.G., and J.N. contributed to plasmid construction and protein purification. J.H., J.X., and L.C. provided technical support. G.M. performed spontaneous mammary tumor induction. B.M. wrote the manuscript. B.M., Y.Z., and Q.C. revised the manuscript with the help of all authors. The project was initiated by Q.C. and Y.Z. and supervised together with B.M.

## Additional information

**Competing interests:** The authors declare no competing interests.

