## [Peer Review File · Nature Communications]

Reviewers' comments:

Reviewer #1 (Remarks to the Author):

This is a very exciting manuscript describing the role of SIAH2 and of NRF in the control of mitochondrial biogenesis in breast cancer. Using a bioinformatic approach, Ma and colleagues identify that mitochondrial biogenesis is reduced in breast cancer and that this is associated with worst prognosis. In the context of the tumor stroma, the biogenesis inversely correlates with the proximity to vessels, suggesting a role for hypoxia in the regulation of the process. Molecularly, this is mediated by the ubiquitination of NRF by SIAH2 and its degradation, leading to metabolic reprogramming.

In complex this is a very strong manuscript. Data are very solid and support the conclusions. The take home message (how metabolism is rewired in cancer by means of mitochondrial biogenesis suppression) is very exciting. The molecular pathway is elucidated to the finest details. I only find Fig. 7 out of context. While it is interesting that the SIAH2-NRF axis has a role on TAM polarization, this constitutes a different story that now is only superficially touched upon (which signal is released in an SIAH2-NRF manner that is required for the polarization? how is it distributed in the tumor bulk? what is its local concentration? and many other questions). I suggest to revise the manuscript by omitting this figure and the related supplementary, as well as the discussion on TAM, focusing the text on the importance of the molecular pathway described here.

Reviewer #2 (Remarks to the Author):

It is well-recognized that mitochondrial function decreases under hypoxia and there is substantial data on the role of HIF1 α regulating this response. New features in this paper are that in hypoxia, NRF1 is degraded via the E3 ligase SIAH2 and a number of NRF1 regulated mitochondrial genes are downregulated as a consequence. This are clear effects on mitochondrial metabolism and this shows a different mechanism compared to that previously reported in regulating mitochondrial biology in hypoxia.

With extensive bioinformatic analysis they show using breast cancer gene array data sets that the mitochondrial genome is downregulated in hypoxia and this correlates with markers of hypoxia biology such as VEGF and Lox. They show that the mechanism is related to proteolysis of NRF1 leading to downregulation of the mitochondrial proteins. NRF1 is not regulated at the RNA level. There are changes in metabolism in glycolysis, oxygen consumption, ATP and NAD/NADH ratios. A further effect was the regulation of a key enzyme of the Krebs cycle, pyruvic dehydrogenase beta, at a transcriptional level. The metabolites produced as a result of the NRF1 degradation appear to regulate macrophage polarisation to M2. Altogether this represents a new insight into how hypoxia regulates mitochondrial metabolism and function contrasting to previous data on HIF1 α -dependent pathways.

However, there are several issues to clarify:

Line 24, 'mitochondrial biogenesis downregulated and positively correlates with outcome'. This should be clarified to say that low expression in breast cancers associated with a poor outcome or aggressive behaviour of the tumour.

There should perhaps be some further references on the role of NRF2, HIF1 α and their mechanisms of action in reducing mitochondrial function, for example, switching of COX subunits to demonstrate the complexity and what is already known about how mitochondria are downregulated in hypoxia.

Line 80, what do they mean by 63% were downregulated and 17% were significantly

downregulated? Surely, only significantly downregulated genes should be counted, that's the whole point of statistics.

Line 93, the gradient of proliferation of mitochondrial marker seems clear here, but there should really be an independent marker of hypoxia, such as pimonidazole or EF5 staining to show where the margin of hypoxia would occur rather than trying to set it arbitrarily by eye. How are these areas related to vasculature and how did they take into account oblique cuts, which give different thicknesses of the rim?

Line 111, they used just two genes to classify if tumours are hypoxic or not. However, there are many more comprehensive hypoxia profiles which can be used and rather than splitting into two groups, it is often much more reliable to split into three groups of equal sizes. Based on the profile I would suggest this is done to bring out the gradient differences. Furthermore, one should normally use the median value because the mean would be effected by outliers. This may actually enhance the analysis.

Line 180, they need to avoid the direct extrapolation that because there is an association of a biomarker with outcome does not mean there is an effect and where this is stated in other sections needs to be removed.

Line 153, the effects of NRF1 knockdown in MD-231 cells are clear. However, they only analysed one other cancer cell type,. To have a general extrapolation to breast cancer from one cell line is not justified. Some of the experiments need to be repeated in a panel of cell lines, even the most simple experiment, which is NRF knockdown and effects on the mitochondrial proteins. In particular, it is important to sub-analyse the clinical data for at least oestrogen receptor positivity, HER2 positivity and triple receptor negative and similarly to use two or three cells lines from each of these just at the very basic analysis to understand the biology appropriately.

Line 172, just to clarify this, do they mean that 63% of 22%, so about 12% of the total genes they analysed?

Line 185, they mention that NRF1 is activated under hypoxia. Could they explain what they mean by this; is it induced at a higher level of protein, is the enzyme activity changed, this needs to be clarified. They elegantly show that the residue K230R is the one essential for the ubiquitination in NRF1 by SIAH2.

Line 237, was the transcription level of SIAH2 related to the hypoxia profile?

Line 249, the metabolic effects are clear with regard to oxygen consumption and mitochondrial mass but the fatty acid data is weak. Low fatty acid levels could be due to decreased uptake, decreased synthesis, increased degradation. In hypoxia, increased uptake and decreased synthesis has been well reported and the information needs to be adequately investigated. They propose that both increased consumption may occur in SIAH2 knockdown but they also say a deficiency in fatty acid synthesis. This needs to be described and the experiments conducted to understand what this means.

They note that PDHB was downregulated at a transcriptional level and they propose it as a potential target gene of NRF1. Since this is such a key enzyme switching between glycolysis and Krebs cycle, I think the actual ChIP analysis needs to be done for the promotor to show that it is indeed an NRF1 responsive gene in this cell line.

Line 303, NRF1 knockdown did not further modify SIAH2 knockdown in tumour growth in vivo but the tumour growth reduction was substantial already and it may be very hard to detect the effect. However, line 301 does not describe the effect of NRF1 knockdown alone at this point. Fig 7a should show the effect of NRF1 shRNA alone.

What was the effect of overexpression of NRF1 wildtype or hypoxia-resistant on the xenograft growth and the biology of the xenografts in terms of vascularisation and Ki67? What was the effect of the NRF1 knockdown? It is interesting the increased necrotic areas which would be expected if the adaptation to hypoxia can no longer be undertaken because of maintenance of mitochondrial function.

The deficiency in angiogenesis could be because hypoxic cells are not surviving and therefore the tissue hypoxia is rather less. Is there any independent measure of hypoxia with pimonidazole or other markers of hypoxia? It could be there is no stimulus to the angiogenesis or less stimulus because of less hypoxic cell mass.

There is a general problem with how some of the statistics are displayed; it is not correct to compare every point in a graph with a preceding baseline, for example, figure 2b, and many other similar figures. Every successive point in a graph is compared. These should be done by analysis of variance or by looking at the shape of the whole curve, not comparing every single point individually. A good example is figure 2a where this is also an inappropriate type of analysis. For many of the graphs, if you do multiple comparisons against one control individually you need to make a correction for the statistics and many of the results would not be significant making that correction although ANOVA or other appropriate statistical analysis would likely show that it was significant taking them all into account.

Figure 5c-the immunochemistry is rather dirty, there is no quantification of the data, it could be that the western blot of the tumours would be more effective in demonstrating a difference. I do appreciate sometimes it is very difficult with antibodies with strong background staining and one has to make do with what is available.

Figure 7. It would be useful to see the xenograft growth curves for the NRF1 knockdown versus NRF overexpression and mutant NRF expression versus the controls. Also to show markers for proliferation and markers for hypoxia e.g. CA9 or pimonidazole to see what happened to the hypoxic cell fraction.

Figure S6c-no stats shown on the graphs.

Reviewer #3 (Remarks to the Author):

The article "Hypoxia orchestrates intratumoral metabolic heterogeneity via regulating mitochondrial biogenesis" by Bio Ma et al. propose a mechanism by which oxygen levels may control mitochondrial biogenesis. While some of the aspects work are already known, their work contributes significantly to our understanding of how metabolic and cellular aspects of the tumor microenvironment are integrated.

Their model proposes that NRF1-dependent mitochondrial biogenesis is inhibited by hypoxia. Under low oxygen levels, the ubiquitin E3 ligase, SIAH2 directly binds to NRF1 and targets it for proteosomal degradation. This mechanism in turn leads to decreased mitochondrial biogenesis and rewiring of cell metabolism with cell autonomous –e.g. changes in glycolytic rates— and non-autonomous effects – such as extracellular lactate-mediated polarization of Tumor Associated Macrophages. I provide below some points that I consider should be addressed before publication.

Major concerns

1) A central aspect of their work is mitochondrial biogenesis and mitochondrial mass. These concepts have very precise and specific definitions which they do not address. For example, they

claim that staining for mitochondrial proteins or measuring expression levels of mitochondria-associated genes, equals to mitochondrial mass, or changes in mitochondrial biogenesis. The statements at the beginning of the paper (figures 1-5) need to be backed up by experiments directly assessing mitochondrial mass such as the ones shown at the end of the paper (eg. in figure 6h).

2) Another central claim of their work is that the mechanism they describe contributes to the spatial heterogeneity of tumors because oxygen levels within tumors are not homogeneous. I really liked this aspect of the work and they should highlight it more and put in the context of previous work on this topic (for example: PDMIDs: 15516961 and 24218566). Having said that, there are some critical experiments that are missing. For example, their experimental evidence is derived from xenografts. These tumors have a very different irrigation nature than spontaneous tumors, which changes the kinetic and spatial properties of metabolite diffusion. In fact, their range of normoxic regions of ~245 μm is about twice as much as the one seen in human and murine tumors (see for example works by Vaupel, Thomlinson or more recent studies of metabolite gradients within tumors). A simple way to address this caveat is to simple staining experiments and look at NRF1, levels, mitochondrial markers, etc. in spontaneous tumors. Ideally proper hypoxia should be labeled in these tumors (using for example pimonidazole or some direct HIF1a targets). Ki67 is not a good marker as hypoxic cells as they can also proliferate depending on their Kras and mTOR status (see for example Palm et al 2015).

3) Why SIAH2 KO increases NRF1 under normoxia (Figure 3h)? This result is not consistent with the absent levels of NRF1 ubiquitination shown in Figure 3i.

4) The distinction of M1 and M2 macrophages in controversial, specially in TAMs. It has been shown that Arg1 is expressed in hypoxic TAMs, while CD206 is expressed by normoxic macrophages located in cortical and perivascular regions. This seems to be differ with their CD206 data. One potential solution to this apparent contraction is to explore CD206/Arg1 expression in different regions of their xenografts (and ideally in spontaneous tumors, as mentioned in point 2).

Minor concerns

- 1) I would seriously consider revising the manuscript language. I think that their writing it is often imprecise and confusing, which may hamper the delivery of their message.
- 2) In the intro (line 61) they seem to give the impression that all their work is bioinformatic.
- 3) There is little or no description of controls for siRNAs experiments (such as multiple guides, often no rescue, etc).
- 4) It is not clear to me why screening shown in Fig3c was done –and worked—under normoxia.
- 5) Figure 3a is not convincing. Differences in blot are tiny and do not seem to match the quantifications in the bars below.
- 6) It is confusing why they used the K230 from figure 2 while they only screened and discover the key properties of this mutation in fig. 4. Please clarify
- 7) In figure 1c and others, 'n' value in bars corresponds to what? Cells? Images?
- 8) Plot in figure 1b is useless.
- 9) It is not clear how they are fitting protein level data (eg. Fig 2a-g). It is apolynomial fit? Why they chose that? Also figure 2h is not very helpful.
- 10) A clear example of use of misleading language: fig 1g prohibitin staining says "Mitochondria". IT IS PROHIBITIN STAINING. This happens all along the paper. There are many ways to directly measure mitochondrial mass, so if you want to claim you measured that, do it. This is even worse in Figure 5c.

Point-by-point response to the referees' comments:

Referee' comments:

Reviewer #1 (Remarks to the Author):

This is a very exciting manuscript describing the role of SIAH2 and of NRF in the control of mitochondrial biogenesis in breast cancer. Using a bioinformatics approach, Ma and colleagues identify that mitochondrial biogenesis is reduced in breast cancer and that this is associated with worst prognosis. In the context of the tumor stroma, the biogenesis inversely correlates with the proximity to vessels, suggesting a role for hypoxia in the regulation of the process. Molecularly, this is mediated by the ubiquitination of NRF by SIAH2 and its degradation, leading to metabolic reprogramming.

In complex this is a very strong manuscript. Data are very solid and support the conclusions. The take home message (how metabolism is rewired in cancer by means of mitochondrial biogenesis suppression) is very exciting. The molecular pathway is elucidated to the finest details. I only find Fig. 7 out of context. While it is interesting that the SIAH2-NRF axis has a role on TAM polarization, this constitutes a different story that now is only superficially touched upon (which signal is released in an SIAH2-NRF manner that is required for the polarization? how is it distributed in the tumor bulk? what is its local concentration? and many other questions). I suggest to revise the manuscript by omitting this figure and the related supplementary, as well as the discussion on TAM, focusing the text on the importance of the molecular pathway described here.

R1: We thank the reviewer for the appreciation of our work and providing suggestions. In xenograft and spontaneous tumor tissues, the polarized M2-TAMs are more abundant in mitochondria-poor regions and are associated with hypoxia (Fig. 7f, h). In the revised manuscript Figure 7f-g, you can see that the degradation of NRF1 is important for the polarization of M2-TAMs and this process is mediated by small molecules. Lactate and PGE2, two mitochondria-related metabolic intermediates, which both have a role in polarizing TAMs, are significantly upregulated under hypoxia. However, their levels are dramatically reduced in NRF1-K230R (hypoxia resistant NRF1 mutant) cells under hypoxia (Fig. 6g-i), which is consistent with the defect of K230R tumors in polarizing TAMs (Fig. 7f).

Reviewer #2 (Remarks to the Author):

It is well-recognized that mitochondrial function decreases under hypoxia and there is substantial data on the role of HIF1 regulating this response. New features in this paper are that in hypoxia, NRF1 is degraded via the E3 ligase SIAH2 and a number of NRF1 regulated mitochondrial genes are downregulated as a consequence. This are clear effects on mitochondrial metabolism and this shows a different mechanism compared to that previously reported in regulating mitochondrial biology in hypoxia.

With extensive bioinformatics analysis they show using breast cancer gene array data sets that the mitochondrial genome is downregulated in hypoxia and this correlates with markers of hypoxia biology such as VEGF and Lox. They show that the mechanism is related to proteolysis of NRF1 leading to downregulation of the mitochondrial proteins. NRF1 is not regulated at the RNA level.

There are changes in metabolism in glycolysis, oxygen consumption, ATP and NAD/NADH ratios. A further effect was the regulation of a key enzyme of the Krebs cycle, pyruvic dehydrogenase beta, at a transcriptional level. The metabolites produced as a result of the NRF1 degradation appear to regulate macrophage polarization to M2. Altogether this represents a new insight into how hypoxia regulates mitochondrial metabolism and function contrasting to previous data on HIF1-dependent pathways.

However, there are several issues to clarify:

Line 24, 'mitochondrial biogenesis downregulated and positively correlates with outcome'. This should be clarified to say that low expression in breast cancers associated with a poor outcome or aggressive behaviour of the tumour.

R2: We thank the reviewer for the appreciation of our work and providing suggestions. We have revised this description in the manuscript.

There should perhaps be some further references on the role of NRF2, HIF1 and their mechanisms of action in reducing mitochondrial function, for example, switching of COX subunits to demonstrate the complexity and what is already known about how mitochondria are downregulated in hypoxia.

R3: We have added more references regarding these in the manuscript. Please see new reference 17-20

Line 80, what do they mean by 63% were downregulated and 17% were significantly downregulated? Surely, only significantly downregulated genes should be counted, that's the whole point of statistics.

R4: Previous Figure 1b was deleted as following the reviewers' suggestions.

Line 93, the gradient of proliferation of mitochondrial marker seems clear here, but there should really be an independent marker of hypoxia, such as pimonidazole or EF5 staining to show where the margin of hypoxia would occur rather than trying to set it arbitrarily by eye. How are these areas related to vasculature and how did they take into account oblique cuts, which give different thicknesses of the rim?

R5: We used GLUT1 as the hypoxia marker and showed that in mouse spontaneous breast tumor tissues, the expression of mitochondrial marker TIMM23 is inversely correlated with tumor hypoxia (Fig 1F).

Line 111, they used just two genes to classify if tumours are hypoxic or not. However, there are many more comprehensive hypoxia profiles which can be used and rather than splitting into two groups, it is often much more reliable to split into three groups of equal sizes. Based on the profile I would suggest this is done to bring out the gradient differences. Furthermore, one should

normally use the median value because the mean would be effected by outliers. This may actually enhance the analysis.

R6: We have added several other well-established hypoxia marker genes *ENO1*, *GLUT1*, *HMOX1* and *PLAUR* together with *VEGF* and *LOX* as the parameters (Fig. 1h). By using their median values, we replotted the graph by analyzing GO enrichment in mitochondrial genes. The reason why we didn't use three groups in surviving curves is because the GO mitochondria enrichment analysis can only be done in comparing two groups and the surviving curve is to indicate the difference of the clinical outcome affected by mitochondrial gene expression in breast cancers.

Line 180, they need to avoid the direct extrapolation that because there is an association of a biomarker with outcome does not mean there is an effect and where this is stated in other sections needs to be removed.

R7: We have revised these statements.

Line 153, the effects of NRF1 knockdown in MD-231 cells are clear. However, they only analysed one other cancer cell type,. To have a general extrapolation to breast cancer from one cell line is not justified. Some of the experiments need to be repeated in a panel of cell lines, even the most simple experiment, which is NRF knockdown and effects on the mitochondrial proteins. In particular, it is important to sub-analyse the clinical data for at least oestrogen receptor positivity, HER2 positivity and triple receptor negative and similarly to use two or three cells lines from each of these just at the very basic analysis to understand the biology appropriately.

R8: In Supplementary Figure 1b, we sub-analyzed the clinical data for HER2, PR and ER positivity, as well as triple negative breast cancers. It seems only HER2 positivity is significantly positive-correlated with mitochondrial marker Prohibitin. We have also used five more breast cancer cell lines (MCF-7, T47D, JIMT-1, MDA-MB-453 and MDA-MB-435) to repeat those NRF1 knockdown experiments and checked the effects on mitochondrial proteins as shown in Figure 2e, which have a consistent result as in MDA-MB-231 cells.

	PR	ER	HER2
MCF-7	+	+	-
T47D	+	+	-
JIMT-1	-	-	+
MDA-MB-453	-	-	+
MDA-MB-435	-	-	-
MDA-MB-231	-	-	-

Line 172, just to clarify this, do they mean that 63% of 22%, so about 12% of the total genes they analysed?

R9: 22% of all known mitochondrial genes are potential NRF1 target genes, and 63% of these potential target genes are downregulated under hypoxia. About 13.86% of total known

mitochondrial genes are downregulated under hypoxia.

Line 185, they mention that NRF1 is activated under hypoxia. Could they explain what they mean by this; is it induced at a higher level of protein, is the enzyme activity changed, this needs to be clarified. They elegantly show that the residue K230R is the one essential for the ubiquitination in NRF1 by SIAH2.

R10: There perhaps a misunderstanding here. We mentioned that the E3 ligases can be activated by hypoxia, not NRF1. Some of these E3 ligases' enzyme activity is enhanced under hypoxia. Corresponding description and reference are added in the manuscript.

Line 237, was the transcription level of SIAH2 related to the hypoxia profile?

R11: We did not see a correlation between the transcription level of SIAH2 and hypoxia profile. However, in breast cancer, SIAH2 is reported amplified at genomic level and is transcriptionally regulated by estrogen, How exactly SIAH2's activity is regulated by hypoxia is not well known. Currently, people know that some hypoxia-driven Kinases like AKT and P38 MAPK could phosphorylate SIAH2 and enhance its E3 ligase activity.

Line 249, the metabolic effects are clear with regard to oxygen consumption and mitochondrial mass but the fatty acid data is weak. Low fatty acid levels could be due to decreased uptake, decreased synthesis, increased degradation. In hypoxia, increased uptake and decreased synthesis has been well reported and the information needs to be adequately investigated. They propose that both increased consumption may occur in SIAH2 knockdown but they also say a deficiency in fatty acid synthesis. This needs to be described and the experiments conducted to understand what this means.

R12: Fatty acid levels were reduced in SIAH2-knockdown cells when cultured under hypoxia (Supplementary Fig. 7g), and this result was further validated in SIAH2-deficient xenograft tumor tissues, which showed weakened Oil Red staining (Supplementary Fig. 7h). These data were consistent with a previous report showing that SIAH2-knockdown cells had a deficiency in fatty acid synthesis. However, after palmitic acid treatment, which gives overdose of fatty acid to exclude the possibility of the influence from newly synthesized fatty acid, SIAH2-knockdown cells also showed reduced Oil Red staining intensity regardless of the oxygen levels (Supplementary Fig. 7i), implying that fatty acid consumption may be also increased in SIAH2-deficient cells.

They note that *PDHB* was downregulated at a transcriptional level and they propose it as a potential target gene of NRF1. Since this is such a key enzyme switching between glycolysis and Krebs cycle, I think the actual ChIP analysis needs to be done for the promotor to show that it is indeed an NRF1 responsive gene in this cell line.

R13: We have done analysis of *PDHB* gene sequence and located a conserved NRF1 binding site within the beginning of the second intron. CHIP-RT analysis further confirms *PDHB* is a NRF1 target gene (Fig 6e, f).

Line 303, NRF1 knockdown did not further modify *SIAH2* knockdown in tumour growth in vivo but the tumour growth reduction was substantial already and it may be very hard to detect the effect. However, line 301 does not describe the effect of NRF1 knockdown alone at this point. Fig 7a should show the effect of NRF1 shRNA alone.

R14: We performed xenograft experiments with wild-type control cells, NRF1 stable knockdown cells and NRF1 stable knockdown cells that stably reconstituted with wild-type NRF1 or the hypoxia-resistant mutant K230R. The results showed that NRF1 knockdown inhibited the tumor growth, whereas reconstituted wild-type NRF1 could completely reverse this growth retardation phenotype (Figure 7a-c). The intact surgically removed K230R tumors showed almost the same volume and weight compare with control groups (Figure 7a-c). However, histological analysis of the xenograft tumor tissues revealed that a slight and a dramatic increase of necrotic areas in wild-type NRF1 and K230R reconstituted tumor tissues respectively (Figure 7d, e).

What was the effect of overexpression of NRF1 wildtype or hypoxia-resistant on the xenograft growth and the biology of the xenografts in terms of vascularisation and Ki67? What was the effect of the NRF1 knockdown? It is interesting the increased necrotic areas which would be expected if the adaptation to hypoxia can no longer be undertaken because of maintenance of mitochondrial function.

The deficiency in angiogenesis could be because hypoxic cells are not surviving and therefore the tissue hypoxia is rather less. Is there any independent measure of hypoxia with pimonidazole or other markers of hypoxia? It could be there is no stimulus to the angiogenesis or less stimulus because of less hypoxic cell mass.

R15: Tumor angiogenesis is a process partially mediated by polarized TAMs. However, our new data showed that the necrosis found within the K230R tumor is due to apoptosis-induced secondary necrosis (Fig. 8a-c). We identified that *FADD* is a novel NRF1 target gene (Fig. 8d-e). Accumulated NRF1 can sustain the expression of *FADD* under hypoxia (Fig. 8f, g), which enhances the susceptibility of tumor cells in response to extrinsic death stimuli, such as TRAIL, and lead to increased apoptosis (Fig 8h, i). Besides, accumulated NRF1 also inhibits the polarization of TAMs, which leads to defects of TAMs in elimination of apoptotic cells. Hence, these two aspects together may cause secondary necrosis and subsequently compromise tumor maintenance. Please see detail data in Figure 7-8.

There is a general problem with how some of the statistics are displayed; it is not correct to compare every point in a graph with a preceding baseline, for example, figure 2b, and many other similar figures. Every successive point in a graph is compared. These should be done by analysis of variance or by looking at the shape of the whole curve, not comparing every single point individually. A good example is figure 2a where this is also an inappropriate type of analysis. For many of the graphs, if you do multiple comparisons against one control individually you need to make a correction for the statistics and many of the results would not be significant making that correction although ANOVA or other appropriate statistical analysis would likely show that is was significant taking them all into account.

R16: Many thanks for pointing out our statistical issues. Those corresponding figures were reanalyzed by variance analysis.

Figure 5c-the immunochemistry is rather dirty, there is no quantification of the data, it could be that the western blot of the tumours would be more effective in demonstrating a difference. I do appreciate sometimes it is very difficult with antibodies with strong background staining and one has to make do with what is available.

R17: Those tissues were all treated with PFA and we did not collect them for WB analysis purposes. However, these xenograft tumor tissues are all derived from those corresponding cells, and we have analyzed them by WB and statistically quantified them in Figure 5a-e.

Figure 7. It would be useful to see the xenograft growth curves for the NRF1 knockdown versus NRF overexpression and mutant NRF expression versus the controls. Also to show markers for proliferation and markers for hypoxia e.g. CA9 or pimonidazole to see what happened to the hypoxic cell fraction.

R18: Please see R14 and detail data in Figure 7

Figure S6c-no stats shown on the graphs.

R19: Statistical data were added.

Reviewer #3 (Remarks to the Author):

The article “Hypoxia orchestrates intratumoral metabolic heterogeneity via regulating mitochondrial biogenesis” by Bio Ma et al. propose a mechanism by which oxygen levels may control mitochondrial biogenesis. While some of the aspects work are already known, their work contributes significantly to our understanding of how metabolic and cellular aspects of the tumor microenvironment are integrated.

Their model proposes that NRF1-dependent mitochondrial biogenesis is inhibited by hypoxia. Under low oxygen levels, the ubiquitin E3 ligase, SIAH2 directly binds to NRF1 and targets it for

proteosomal degradation. This mechanism in turn leads to decreased mitochondrial biogenesis and rewiring of cell metabolism with cell autonomous –e.g. changes in glycolytic rates— and non-autonomous effects – such as extracellular lactate-mediated polarization of Tumor Associated Macrophages. I provide below some points that I consider should be addressed before publication.

Major concerns

1) A central aspect of their work is mitochondrial biogenesis and mitochondrial mass. These concepts have very precise and specific definitions which they do not address. For example, they claim that staining for mitochondrial proteins or measuring expression levels of mitochondria-associated genes, equals to mitochondrial mass, or changes in mitochondrial biogenesis. The statements at the beginning of the paper (figures 1-5) need to be backed up by experiments directly assessing mitochondrial mass such as the ones shown at the end of the paper (eg. in figure 6h).

R20: We thank the reviewer for the appreciation of our work and providing suggestions. In the revised manuscript, now we use nuclear-encoded mitochondrial genes (NEMGs) instead. Because hypoxia could alter the mitochondrial structures and the composition of mitochondrial lipids and proteins, however, the principle of mitotracker or FAO probe detect mitochondrial mass is based on those factors. It may not accurate to compare the results between normoxia and hypoxia. So we just compare different modified cells at same conditions by FACs and compare protein and their corresponding mRNAs' levels by WB and RT-PCR in between normoxia- and hypoxia-treated cells.

2) Another central claim of their work is that the mechanism they describe contributes to the spatial heterogeneity of tumors because oxygen levels within tumors are not homogeneous. I really liked this aspect of the work and they should highlight it more and put in the context of previous work on this topic (for example: PDMIDs: 15516961 and 24218566). Having said that, there are some critical experiments that are missing. For example, their experimental evidence is derived from xenografts. These tumors have a very different irrigation nature than spontaneous tumors, which changes the kinetic and spatial properties of metabolite diffusion. In fact, their range of normoxic regions of ~245 um is about twice as much as the one seen in human and murine tumors (see for example works by Vaupel, Thomlinson or more recent studies of metabolite gradients within tumors). A simple way to address this caveat is to simple staining experiments and look at NRF1, levels, mitochondrial markers, etc. in spontaneous tumors. Ideally proper hypoxia should be labeled in these tumors (using for example pimonidazole or some direct HIF1a targets). Ki67 is not a good marker as hypoxic cells as they can also proliferate depending on their Kras and mTOR status (see for example Palm et al 2015).

R21: Many thanks for the reviewer's appreciation of our work. We have added more discussion in the text and also provided a new molecular mechanism to explain the consequence if NRF1 is not spatially regulated (Figure 8). We also used Glut1 as the hypoxia marker and tested the relationship among polarized TAMs, mitochondria and hypoxia. In mouse spontaneous breast

tumor tissues, we found that mitochondria are inversely correlated with hypoxia marker (Fig 1f, Fig 7h). The ARG1+ polarized M2-TAMs were consistently enriched in hypoxic and also mitochondria-pool regions (Fig. 7f, h).

3) Why SIAH2 KO increases NRF1 under normoxia (Figure 3h)? This result is not consistent with the absent levels of NRF1 ubiquitination shown in Figure 3i.

R22: SIAH2 is known to promote itself ubiquitination and degradation under normoxia, and this process could be reversed by hypoxia, which promotes the binding with its substrates. Even though SIAH2 is more activated under hypoxia, it still has E3 activity under normoxia. In Figure 3i, if the exposure time is long enough, we should see the smear band of polyubiquitinated NRF1.

4) The distinction of M1 and M2 macrophages is controversial, especially in TAMs. It has been shown that Arg1 is expressed in hypoxic TAMs, while CD206 is expressed by normoxic macrophages located in cortical and perivascular regions. This seems to differ with their CD206 data. One potential solution to this apparent contradiction is to explore CD206/Arg1 expression in different regions of their xenografts (and ideally in spontaneous tumors, as mentioned in point 2).

R23: We consistently used ARG1 as the M2-TAMs' marker and tested that in spontaneous tumor tissues. Please see R21.

Minor concerns

1) I would seriously consider revising the manuscript language. I think that their writing is often imprecise and confusing, which may hamper the delivery of their message.

R24: We have improved our manuscript.

2) In the intro (line 61) they seem to give the impression that all their work is bioinformatic.

R25: This perhaps is misinterpretation. We try not to and improve it.

3) There is little or no description of controls for siRNAs experiments (such as multiple guides, often no rescue, etc).

R26: The siRNAs were bought from RiboBio. We have tested the specificity by using three different siRNAs against target genes. We choose the best one with high efficiency of knockdown. Control groups were transfected with negative control siRNA, RiboBio siN05815122147. Detail information were shown in the Methods. For shRNA knockdown, we have WT-NRF1 and K230R to rescue the phenotypes.

4) It is not clear to me why screening shown in Fig3c was done –and worked—under normoxia.

R27: Those indicated E3 ligases were previously reported involved in hypoxia response. Taking SIAH2 for example, SIAH2 is known to promote itself ubiquitination and degradation under normoxia, and this process could be reversed by hypoxia, which promotes the binding with its substrates. Even though SIAH2 is more activated under hypoxia, it still has E3 activity under normoxia. In Figure 3c. NRF1 and those E3 ligases were co-transfected, which makes it easier to detect which E3 ligase could degrade NRF1. As you can also see in Supplementary Figure 4, overexpression of SIAH2 can mediate the polyubiquitination and degradation of co-transfected NRF1.

5) Figure 3a is not convincing. Differences in blot are tiny and do not seem to match the quantifications in the bars below.

R28: The experiments were repeated so many times. We changed a clear one in Figure 3a.

6) It is confusing why they used the K230 from figure 2 while they only screened and discover the key properties of this mutation in fig. 4. Please clarify

R29: Actually we identified K230 first and then tested its effect on mitochondrial proteins. In Figure 2, we initially thought it may be more informative to compare NRF1 knockdown cells and those cells rescued by WT or K230R NRF1, so we put some data forward. Since this confuses people, we have made rearrangements and put some of those data from Figure 2 to Figure 5.

7) In figure 1c and others, 'n' value in bars corresponds to what? Cells? Images?

R30: In Figure 1c-e, n= samples from patient

8) Plot in figure 1b is useless.

R31: We deleted this figure.

9) It is not clear how they are fitting protein level data (eg. Fig 2a-g). It is apolynomial fit? Why they chose that? Also figure 2h is not very helpful.

R32: We deleted figure 2h. The protein levels were quantified and analyzed by variance analysis. Each plot is the mean value of relative expression level of the corresponding protein. The graph is to show the shape of the whole curve.

10) A clear example of use of misleading language: fig 1g prohibitin staining says "Mitochondria". IT IS PROHIBITIN STAINING. This happens all along the paper. There are many ways to directly measure mitochondrial mass, so if you want to claim you measured that, do it. This is even worse in Figure 5c.

R33: We have improved this and clearly showed what we stained and measured in the revised manuscript.

REVIEWERS' COMMENTS:

Reviewer #1 (Remarks to the Author):

Authors have provided new data to explain the observed TAM phenotype and I am satisfied with them. I believe that the paper is a very strong candidate for publication.

Reviewer #2 (Remarks to the Author):

You ave answered my main concerns in the new work

Reviewer #3 (Remarks to the Author):

The article "The SIAH2-NRF1/alpha-Pal axis spatially regulates tumor microenvironment remodeling for tumor progression" by Bio Ma et al. propose a mechanism by which oxygen levels may control mitochondrial biogenesis. This revised manuscript is an improvement from the previous submission. I will revise the points raised in the previous submission and some other points that should be addressed before publication.

1) Major concern on first revision → "A central aspect of their work is mitochondrial biogenesis and mitochondrial mass..."

This point has been mostly addressed. I would recommend however to not use the term mitochondrial mass in Fig S1 for example where there is no mass measurements. While mitochondrial proteins often may correlate with mitochondrial mass, they are not the same thing.

2) Major concern on first revision → "Another central claim of their work is that the mechanism they describe contributes to the spatial heterogeneity of tumors because oxygen levels within tumors are not homogeneous..."

This point has been well addressed. Interesting new data shown in Fig 8.

3) Major concern on first revision → "Why SIAH2 KO increases NRF1 under normoxia (Figure 3h)? This result is not consistent with the absent levels of NRF1 ubiquitination shown in Figure 3i."

This point has been addressed.

4) Major concern on first revision → "The distinction of M1 and M2 macrophages in controversial, specially in TAMs."

This point has been addressed.

Minor concerns from first revision

1) I would seriously consider revising the manuscript language. I think that their writing it is often imprecise and confusing, which may hamper the delivery of their message.

I would still recommend revision the language.

2) In the intro (line 61) they seem to give the impression that all their work is bioinformatic.

Addressed.

3) There is little or no description of controls for siRNAs experiments (such as multiple guides, often no rescue, etc).

Addressed.

4) It is not clear to me why screening shown in Fig3c was done –and worked—under normoxia.

Addressed.

5) Figure 3a is not convincing. Differences in blot are tiny and do not seem to match the quantifications in the bars below.

Addressed.

6) It is confusing why they used the K230 from figure 2 while they only screened and discover the key properties of this mutation in fig. 4. Please clarify

Addressed.

7) In figure 1c and others, 'n' value in bars corresponds to what? Cells? Images?

Addressed.

8) Plot in figure 1b is useless.

Addressed.

9) It is not clear how they are fitting protein level data (eg. Fig 2a-g). It is a polynomial fit? Why they chose that? Also figure 2h is not very helpful.

Not addressed. Please explain in the legend or methods what the lines are. Are they a fit?

Alternatively, I would recommend deleting the lines since they don't add much.

10) A clear example of use of misleading language: fig 1g prohibitin staining says "Mitochondria". IT IS PROHIBITIN STAINING. This happens all along the paper. There are many ways to directly measure mitochondrial mass, so if you want to claim you measured that, do it. This is even worse in Figure 5c.

Addressed.

Point-by-point response to the referees' comments

Reviewer #1 (Remarks to the Author):

Authors have provided new data to explain the observed TAM phenotype and I am satisfied with them. I believe that the paper is a very strong candidate for publication.

Reviewer #2 (Remarks to the Author):

You ave answered my main concerns in the new work

Reviewer #3 (Remarks to the Author):

The article “The SIAH2-NRF1/alpha-Pal axis spatially regulates tumor microenvironment remodeling for tumor progression” by Bio Ma et al. propose a mechanism by which oxygen levels may control mitochondrial biogenesis. This revised manuscript is an improvement from the previous submission. I will revise the points raised in the previous submission and some other points that should be addressed before publication.

1) Major concern on first revision → “A central aspect of their work is mitochondrial biogenesis and mitochondrial mass...”

This point has been mostly addressed. I would recommend however to not use the term mitochondrial mass in Fig S1 for example where there is no mass measurements. While mitochondrial proteins often may correlate with mitochondrial mass, they are not the same thing.

R: We use Prohibitin instead of using mitochondrial mass.

2) Major concern on first revision → “Another central claim of their work is that the mechanism they describe contributes to the spatial heterogeneity of tumors because oxygen levels within tumors are not homogeneous...”

This point has been well addressed. Interesting new data shown in Fig 8.

3) Major concern on first revision → “Why SIAH2 KO increases NRF1 under normoxia (Figure 3h)? This result is not consistent with the absent levels of NRF1 ubiquitination shown in Figure 3i.”

This point has been addressed.

4) Major concern on first revision → “The distinction of M1 and M2 macrophages in controversial, specially in TAMs.”

This point has been addressed.

Minor concerns from first revision

1) I would seriously consider revising the manuscript language. I think that their writing it is often imprecise and confusing, which may hamper the delivery of their message.

I would still recommend revision the language.

2) In the intro (line 61) they seem to give the impression that all their work is bioinformatic.

Addressed.

3) There is little or no description of controls for siRNAs experiments (such as multiple guides, often no rescue, etc).

Addressed.

4) It is not clear to me why screening shown in Fig3c was done –and worked—under normoxia.

Addressed.

5) Figure 3a is not convincing. Differences in blot are tiny and do not seem to match the quantifications in the bars below.

Addressed.

6) It is confusing why they used the K230 from figure 2 while they only screened and discover the key properties of this mutation in fig. 4. Please clarify

Addressed.

7) In figure 1c and others, ‘n’ value in bars corresponds to what? Cells? Images?

Addressed.

8) Plot in figure 1b is useless.

Addressed.

9) It is not clear how they are fitting protein level data (eg. Fig 2a-g). It is a polynomial fit? Why they chose that? Also figure 2h is not very helpful.

Not addressed. Please explain in the legend or methods what the lines are. Are they a fit? Alternatively, I would recommend deleting the lines since they don't add much.

R: We deleted the lines.

10) A clear example of use of misleading language: fig 1g prohibitin staining says “Mitochondria”. IT IS PROHIBITIN STAINING. This happens all along the paper. There are many ways to directly measure mitochondrial mass, so if you want to claim you measured that, do it. This is even worse in Figure 5c.

Addressed.